# Equilibrium distance from long-range dune interactions

Jean Vérité[1], Clément Narteau[1], Olivier Rozier[1], Jeanne Alkalla[1], Laurie Barrier[1], and Sylvain Courrech du Pont[2]

[1]Université Paris Cité, Institut de Physique du Globe de Paris, CNRS, Paris, France.
[2]Laboratoire Matière et Système Complexes, Université de Paris, CNRS, Paris, France.

**Correspondence:** Jean Vérité (verite@ipgp.fr)

**Abstract.** Flow perturbations induced by dune topography affect sediment transport locally, but can also be felt over long distances altering the dynamics of isolated neighbouring dunes downstream. In order to work under optimal conditions that eliminate transverse flow components, collisions and mass exchange between dunes, we study here these long-range interactions using a 2D numerical model where two equal-sized dunes lying on a non-erodible bed are exposed to a symmetric reversing flow. Depending on the initial spacing, dunes either attract or repel each other, to eventually converge towards a steady-state spacing. This equilibrium distance decreases with flow strength and increases with period of flow reorientation and dune size. It is mainly controlled by the reversing dune shape and the structure of the turbulent wake it generates, which continuously modulates the mean shear stress on the downstream dune. Under multi-directional wind regimes, these long-range flow perturbations offer an alternative mechanism for wavelength selection in linear dune fields with non-erodible interdune areas. Within these dune fields, estimates of mean shear stress could be used to assess the relative migration rate and the state of attraction/repulsion between neighbouring dunes.

## 1 Introduction

While the development of the largest dune systems and giant dunes are still debated given the various conditions that contribute to their formation over time (Andreotti et al., 2009; Gao et al., 2015b; Gunn et al., 2022), the different elementary dune types at smaller scale can be rationally linked to wind flow and sand availability (Wasson and Hyde, 1983; Courrech du Pont et al., 2024). The reasoning is built around the permanent feedback between topography, bed shear stress, and sediment transport, which has been formalised to provide a comprehensive description of the dune instability (e.g., Kennedy, 1963; Lü et al., 2021). Whatever the wind regime, where the sand availability is not limited, this instability results in the emergence of a dune pattern with a characteristic wavelength and the continuous alternation between zones of deposition and erosion (e.g., Elbelrhiti et al., 2005; Gadal et al., 2019). However, in areas of low sand availability, the dune instability is lost and wavelength coarsening can no longer be considered to explain the formation of periodic dune fields (Gadal et al., 2020). Under these conditions, isolated dunes separated by sediment-free surfaces can grow in height, migrate or elongate (Courrech du Pont et al., 2014; Rozier et al., 2019; Lü et al., 2022; Courrech du Pont et al., 2024), to eventually produce large scale dune fields that continuously adapt to wind climate (e.g., Myrow et al., 2018; Gunn, 2023). Then, one particularly poorly researched area is the origin of the characteristic wavelength and the manner in which the dune topography can perturb the flow over long distances.

Topography-induced flow perturbations are now supported by a growing body of research, which demonstrates that an upstream dune modifies the shape and migration rate of downstream dunes by perturbing the bed shear stress over long distances in its wake. The structure of turbulent flow over dunes has been analysed by numerous laboratory experiments (Bennett and Best, 1995; Frank and Kocurek, 1996; Walker and Nickling, 2002; Dong et al., 2008; Palmer et al., 2012; Bristow et al., 2019, 2021; Cai et al., 2021), numerical simulations (Stoesser et al., 2008; Omidyeganeh et al., 2013; Anderson and Chamecki, 2014; Smith et al., 2017; Wang et al., 2017; Wang and Anderson, 2018; Wang et al., 2019; Jackson et al., 2020; Love et al., 2022), and field measurements (Arens et al., 1995; Neuman et al., 1997; Baddock et al., 2007; Walker et al., 2009; Wiggs and Weaver, 2012). All sources describe an acceleration on the stoss side, and the formation of a recirculation bubble in the lee side of dunes when slopes are sufficiently steep. This results from the development of a shear layer at the dune crest, which eventually leads to flow separation, and a reattachment point a few dune heights away. Secondary flow structures that develop from this shear layer enhance the turbulence intensity above the recirculation zone and downstream of the reattachment point. These abrupt variations in shear stress in the wake of a dune modify the potential rate of sediment transport. However, the impact of such a turbulent shear zone on the morphodynamics of downstream dunes has received less attention because most of these detailed studies of fluid dynamics have been carried out on fixed geometries or bedforms with a well-established wavelength. Instead, another set of numerical simulations (Eastwood et al., 2011; Katsuki et al., 2011; Zhou et al., 2019; Jarvis et al., 2023) and subaqueous laboratory experiments (Endo et al., 2004; Katsuki et al., 2005; Groh et al., 2009; Assis and Franklin, 2020, 2021; Jarvis et al., 2022; He et al., 2023), focusing on how collisions play an important role in dune pattern coarsening, indirectly highlights the long-range flow-induced interactions between dunes separated by non-erodible beds.

Although the ejection mechanism during barchan collision is surely impacted by the exchange of mass between the two dunes (Elbelrhiti, 2012), it has been typically described as the result of long-range flow-induced interactions. Under unimodal flow regime, when a smaller and therefore faster barchan collides with a larger one, laboratory experiments (Hersen et al., 2004; Hersen and Douady, 2005) and field observations (Bourke, 2010) show that the impacted dune undergoes greater shear and erosion as the impacting dune approaches. This significantly changes dune interactions, as whole sections of dunes that have not yet come into contact are likely to repel each other, favoring the ejection of a smaller barchan from the impacted one (Fig. 1a). This repulsion mechanism appears all the more obvious in a narrow circular flume, in which there is no flow channelling effect between dunes or horizontal flow deflection around dunes (Bacik et al., 2020, 2021a, b). In these 2D experiments, bedforms can be described as isolated transverse dunes, and there is evidence for a flow-induced repulsion mechanism, preventing collisions and allowing the dunes to maintain a steady spacing (Fig. 1b). In another flume experiment in which multi-directional flow regimes can be explored (Courrech du Pont et al., 2014), these long-range interactions also exist for linear dunes when elongation is the prevailing growth mechanism (Figs. 1c-d). At the edge of the experimental dune field, where the flow is only perturbed in one direction by the topography, isolated dunes migrate laterally in a direction transverse to that of the resultant transport. Within the dune field, newly formed dunes also increase the spacing between surrounding dunes as they start to elongate. It eventually produces a periodic dune pattern with a wavelength which does not result from the dune instability (Gadal et al., 2020). Together, all these experiments demonstrate that dunes located in the wake of other dunes have specific dynamics that need to be investigated from the perspective of the flow perturbation over long distance. This of particular

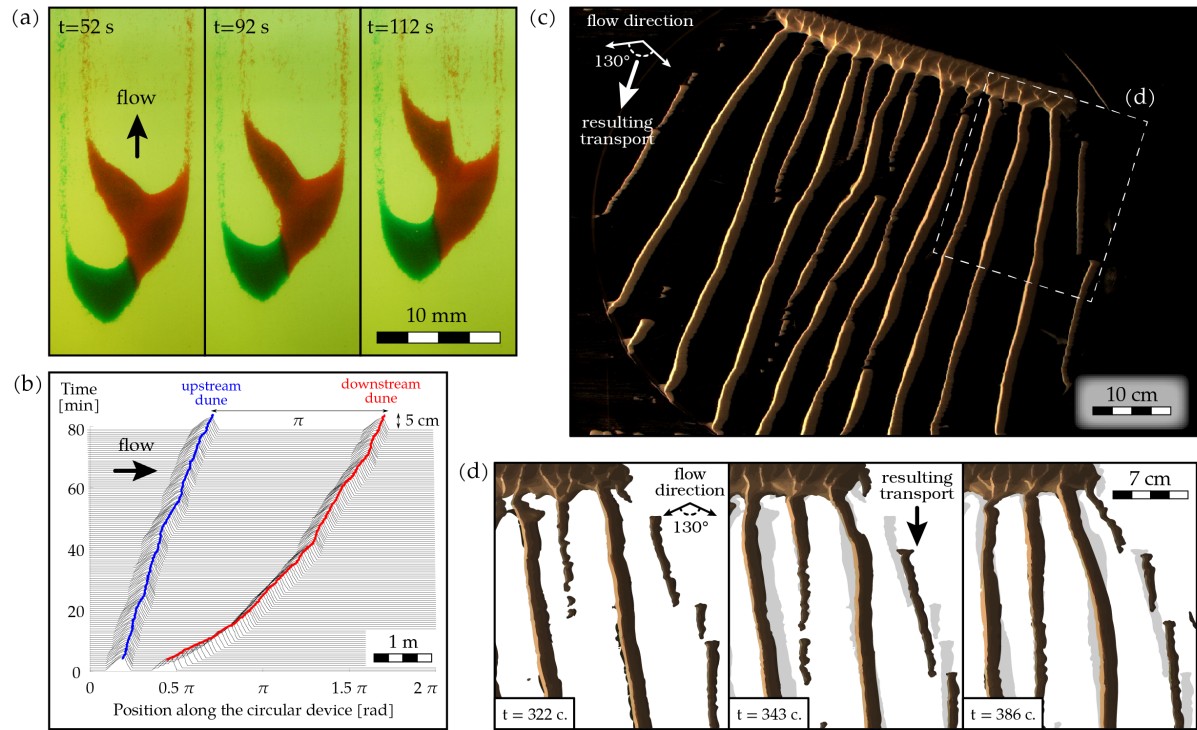

**Figure 1. Flow-induced long-range interactions in subaqueous dune experiments. (a)** Collision of barchans under unidirectional flow. As the impacting dune (green) approaches, a whole section of the impacted dune (red) is sheared and ejected from the main dune body; image credit: Hersen and Douady (2005). **(b)** Space-time diagram of transverse dunes interacting over a large distance in a narrow circular flume. While the upstream dune migrates at constant rate, the downstream dune accelerates then decelerates, i.e., repulsing each over then stabilising at an antipodal configuration (modified from an initial figure of Bacik et al. (2020)). **(c)** Development of longitudinal linear dunes from a linear sand source under a symmetric bidirectional flow regime with a divergence angle of $144^{\rm o}$. **(d)** Three snapshots illustrating the evolution of linear dunes in flume experiments over time, after 322, 343 and 386 flow cycles. Dunes have been extracted from the images using a color segmentation. Grey areas show the previous location of dunes. When a new isolated linear dune elongates from the source in the resulting transport direction, pre-existing surrounding dunes are progressively pushed away, illustrating the repulsion phenomenon, which also affects isolated dunes ejected on the right side.

significance because differences in migration and/or elongation rates could control the overall properties of dune fields in zones of low sand availability.

Numerical models offer the opportunity to quantify the dynamic interactions between topography, shear stress and sediment transport. Nevertheless, they are often time-consuming or built from a simplification of the flow over a complex topography. As a compromise, we choose here to use a cellular automaton approach that couples models of sediment transport and turbulent flow (Narteau et al., 2009; Rozier and Narteau, 2014). Although these two components of the same model lack the sophistication of advanced computational methods for simulating the details of turbulent structures and grain transport properties,

65

their couplings result in a self-organised system that reproduce a wide range of dune types. All these emergent dune features, including superimposed bedforms (Zhang et al., 2010), star dunes (Zhang et al., 2012), barchans (Zhang et al., 2014; Lin et al., 2024) or elongating dunes (Rozier et al., 2019), can then be analysed in detail depending on flow strength and direction, based on the dynamic interactions that arise spontaneously from the coupling between the fluid and sedimentary compartments.

To eliminate the contribution of collisions, mass exchanges, lateral sediment transport and flow channelling effect between neighbouring dunes, the model is used in 2D with a pair of isolated dunes evolving on a non-erodible ground under symmetric reversing flow regimes (Section 2). Although these simulations do not allow to reproduce the full range of turbulent flows and sediment transport processes leading to pattern coarsening in 3D (Endo et al., 2004; Katsuki et al., 2005; Assis and Franklin, 2020, 2021; Lima et al., 2024), they consistently reproduce turbulent recirculation zones (Herrmann et al., 2005; Michelsen et al., 2015; Araújo et al., 2013) and dune interactions (Coleman and Melville, 1994; Diniega et al., 2010; Gao et al., 2015; Bacik et al., 2020, 2021a; Jarvis et al., 2022, 2023), while reducing the computation cost and simplifying the analysis of the dynamic interactions between dune topography, shear stress and sediment transport. Due to symmetric reversing flow regimes, dunes are alternately located in the wake of the other, they do not exchange sediment and act as sediment traps. Since dunes move back and forth according to the two flows, theoretically with a zero resultant migration rate, any systematic deviations of dune spacing result from long-range flow-induced perturbations. Here we study these long-range interactions by measuring the mean shear stress on dune stoss slopes, to show how it governs the relative migration rate of dunes (Section 3). Thus, we can estimate how dune shape and crest reversal dynamics affect the flow and the attraction/repulsion state between two neighbouring dunes (Section 4).

## 2 Methods

### 2.1 ReSCAL dune model

The ReSCAL dune model combines two cellular automata simulating sediment transport and turbulent flow (Narteau et al., 2009; Rozier and Narteau, 2014). Although these two cellular automata can operate independently, the model's strength lies in their couplings, which introduce the permanent feedback of an evolving topography on the flow, and the dependence of sediment transport rate on flow strength.

#### 2.1.1 Model of sediment transport

The physical environment is fully described by a regular lattice of square (2D) or cubic (3D) cells with an elementary length scale $l_0$. Cells can be in one of the three states: fluid, mobile, and immobile sediment. Individual physical processes are associated with different sets of transitions within pairs of nearest-neighbour cells, and a characteristic time scale expressed in units of $t_0$. Considering these time scales, all transitions are incorporated into a continuous-time stochastic process, with a transition probability per unit of time for each of them (Narteau et al., 2001). The most important are the transition rates for erosion, $\Lambda_e$ (immobile $\rightarrow$ mobile sediment), deposition, $\Lambda_c$ (mobile $\rightarrow$ immobile sediment), and transport, $\Lambda_t$ (motion of

mobile sediment). To take into account avalanche processes, we impose a repose angle $\theta_\mathrm{c} = 35°$ for the granular material. To reduce artefacts related to the square lattice and produce realistic slip faces in the model, the slope is first roughly estimated using the elevation of sedimentary cells at distances of $\pm 2\,l_0$, then recalculated using a linear interpolation of the elevation on four nearest neighbors (see more details in Zhang et al. (2014) and in the Supplementary material S1 of Gao et al. (2016)).

### 2.1.2 Model of fluid dynamics

To simulate fluid flow, we implement a lattice gas cellular automaton that converts discrete motions of a finite number of fluid particles into physically meaningful quantities (Frisch et al., 1986). Fluid particles are vertically confined between the sediment layer, lying on flat bedrock, and a flat ceiling. This confinement prevents dissipation of momentum by keeping the number of fluid particles constant. At each iteration, particles can move from cell to cell along the direction specified by their velocity vectors. Depending on the configuration of fluid particles at each point, the collision dynamics can modify the velocity vectors of each particle. The repetition of propagation and collision phases generates fluxes of particles that can be converted into a fluid velocity field $\boldsymbol{u}$ by averaging velocity vectors over space and time at the elementary length scale $l_0$ of the model of sediment transport.

### 2.1.3 Coupling between topography and flow

At the surface of the sediment layer, we consider a no-slip boundary condition: all fluid particles that collide with sedimentary cells bounce back along their incident direction. Thus, the velocity field is null at the sediment-fluid interface and there is a permanent feedback of the topography on the fluid flow. As a result, this model spontaneously produces an increase in flow velocity on the stoss side of dunes and a flow recirculation on their lee side (Zhang et al., 2010, 2012). It also generates a logarithmic vertical velocity profile from which it is possible to derive the bed shear stress,

$$\tau_\mathrm{s} = \tau_0 \frac{\partial \boldsymbol{u}}{\partial \boldsymbol{n}}, \tag{1}$$

where $\boldsymbol{n}$ is the normal to the topography and $\tau_0$ is the stress scale of the model.

### 2.1.4 Coupling between flow and sediment transport

Additionally, the flow continuously modifies the topography through sediment transport where the bed shear stress is high enough to mobilize sediment cells. In practice, we consider that the erosion rate $\Lambda_e$ is linearly related to the bed shear stress $\tau_\mathrm{s}$ according to

$$\Lambda_e(\tau_\mathrm{s}) = \Lambda_0 \begin{cases} 0 & \text{for } \tau_\mathrm{s} \leq \tau_1, \\ \dfrac{\tau_\mathrm{s} - \tau_1}{\tau_2 - \tau_1} & \text{for } \tau_1 \leq \tau_\mathrm{s} \leq \tau_2, \\ 1 & \text{else,} \end{cases} \tag{2}$$

where $\Lambda_0$ is a constant rate, $\tau_1$ is the threshold shear stress for motion inception and $\tau_2$ is a parameter to adjust the slope of the linear relationship. For consistency, we have $\tau_2 \gg \tau_s$ and $(\tau_2 - \tau_1)/\tau_0 = 100$. By definition, $(\tau_s - \tau_1)$ is the excess shear stress from which we can derive an equivalent flow strength. As the erosion rate, $\Lambda_e$, is continuously decreasing with an increasing $\tau_1$-value, the corresponding decay of the saturated flux on a flat sand bed is associated with a decreasing flow strength (Narteau et al., 2009).

Given the dynamic interactions between topography, fluid flow and sediment transport in the model, the aspect ratio of the elementary cells is an independent variable that should have no influence on dune morphodynamics. In fact, it depends entirely on the dependence of the transport rate on the shear stress distribution (Narteau et al., 2009; Zhang et al., 2010; Rozier et al., 2019).

### 2.1.5 Length and time scales of the model

The elementary length and time scales, $\{l_0, t_0\}$, and the threshold shear stress $\tau_1$ are entirely defined with respect to the dune instability using the most unstable wavelength, the mean flow strength and the corresponding saturated sand flux (Narteau et al., 2009). The model can then be used in all types of physical environments where the dune instability is observed to provide quantitative estimates of the evolution of dune fields (Lü et al., 2021). For instance, the same simulation with a given $\tau_1$-value can be used to investigate dunes in laboratory experiments conducted underwater (Jarvis et al., 2022, 2023) or in the field (Lucas et al., 2015; Lü et al., 2017), taking into account the corresponding $\{l_0, t_0\}$-values. For example, $\tau_1/\tau_0 = 20$, $l_0 = 0.5$ m and $t_0 = 8.0 \times 10^{-4}$ yr are typical parameters for terrestrial aeolian dunes, while $l_0 = 5 \times 10^{-4}$ m and $t_0 = 1.6 \times 10^{-10}$ yr for terrestrial subaqueous dunes (Zhang et al., 2014). Theoretically, this rescaling strategy also permits the model to overcome the fundamental limitation related to the arbitrary choice of the elementary length scale inherent to a cellular automaton approach.

### 2.2 Initial conditions, model setup and outputs

Numerical simulations are run using a 2D periodic domain, $100\,l_0$ in height and $1000\,l_0$ in length. Two triangular sand piles of identical size $S$, at the repose angle $\theta_c$, are placed at an equal distance from the center of the domain on a flat non-erodible bed. Except for the simulations shown in Figures 2a,b, we assume a symmetric bidirectional flow regime with a period of flow reorientation $\Delta T$. During each flow period, two flows of the same strength and duration, $\Delta T/2$, blow alternately, so that there is zero resultant transport on the non-erodible bed away from any topography. After each flow reversal, we stabilize the flow over the current dune topography by implementing $10^4$ iterations of the lattice-gas cellular automaton model when the transport model is still inactive. Thus, a statistically steady turbulent state is reached at the beginning of each flow period. Using different random seeds, all simulations were repeated 10 times to ensure reproducibility, and to estimate the variability of our results due to the non-deterministic nature of the model.

We investigate the influence of the period of flow reorientation, $10 \leq \Delta T \leq 10^5$, dune size, $S/l_0^2 = \{2000, 1500, 1000\}$, and threshold shear stress value, $\tau_1/\tau_0 = \{0, 10, 20\}$. In order to prevent collisions and mass exchanges that occur when dunes are in close proximity to one another (Jarvis et al., 2022, 2023), the initial spacing between dunes are always larger than $120\,l_0$, $130\,l_0$ and $140\,l_0$ for dune size $S/l_0^2 = 1000, 1500$ and $2000$, respectively. The dune turnover time $T_D = S/Q_{\text{crest}}(\tau_1)$ is the

time for a dune to remobilise all its sediment, where $Q_{\text{crest}}(\tau_1)$ is the flux at the crest for a given threshold shear stress value $\tau_1$. We save the topography built by the immobile sedimentary cells at the end of each flow period to measure the length, $L_{\text{D}}$, and height, $H$, of the dunes, as well as the length of their stoss slope, $L$, and the interdune spacing, $\lambda_{\text{D}}$ (i.e., distance between the two centres of mass). These variables are used to compute the dune aspect ratio, $H/L$, and shape ratio, $S/(L_{\text{D}}H)$. Shape ratios of $0.5$ and $0.66$ correspond to triangular and parabolic dune shapes, respectively.

From the bed shear stress measured on the dune, we compute the mean shear stress,

$$\langle \tau_{\text{s}} \rangle_{\text{D}} = \frac{1}{L} \int\limits_{x_{\text{t}}}^{x_{\text{t}}+L} \tau_{\text{s}}(x)\mathrm{d}x, \tag{3}$$

where $x_{\text{t}}$ is the position of the dune toe. This integral estimates the overall transport on the stoss slope, and does not take into account the lee side of the dune where the avalanche processes prevail. The difference in mean shear stress between the upstream and downstream isolated dunes,

$$\Delta \langle \widehat{\tau_{\text{s}}} \rangle = \frac{\langle \tau_{\text{s}} \rangle_{\text{D}1} - \langle \tau_{\text{s}} \rangle_{\text{D}2}}{\langle \tau_{\text{flat}} \rangle}, \tag{4}$$

is used as a proxy for the relative migration rate between these two dunes. In Equation 4, $\langle \tau_{\text{flat}} \rangle$ is the mean basal shear stress averaged over $150\, l_0$ on a flat sand bed away from any topography.

## 3 Results

### 3.1 Attraction and repulsion of isolated dunes

In the model, when two identical dunes are exposed to a unidirectional flow of constant strength, Figures 2a-b show that their short-term dynamics depend on their initial spacing, $\lambda_{\text{I}}$. While the upstream dune has a constant migration rate, the downstream dunes migrate faster ($+3\%$) or slower ($-1.5\%$) for small or large initial spacings, respectively. Therefore, a wake-induced repulsion or attraction mechanism operates initially and then slowly disappears as the dune spacing evolves. Regardless of the initial spacing, as long as it exceeds the threshold value that prevents dune collisions, dune pairs eventually reach the same equilibrium distance, where both dunes migrate at the same rate as a single dune under identical flow conditions.

To focus on the relative migration rate between dunes, we apply a symmetric reversing flow to the same initial dune configurations. Under these conditions, dunes are alternately located in the wake of the other dune and their resultant migration rate should be zero, except as a result of stochastic fluctuations. Thus, the evolution of the distance between dunes directly gives the relative migration rate resulting from their long-range and flow-induced interactions. Again, there is a transition from repulsion to attraction as the initial dune spacing increases and a long-term equilibrium distance is reached (Figs. 2c-f). Despite crest reversals, this distance is of the same order of magnitude as the one observed for a unidirectional flow of the same strength.

Under both unidirectional and bidirectional flow regimes, simulations in Figure 2 show the repulsion or attraction between dunes before they stabilise at an equilibrium distance, $\lambda_{\text{D}}$. Since this equilibrium distance appears to be controlled by flow-

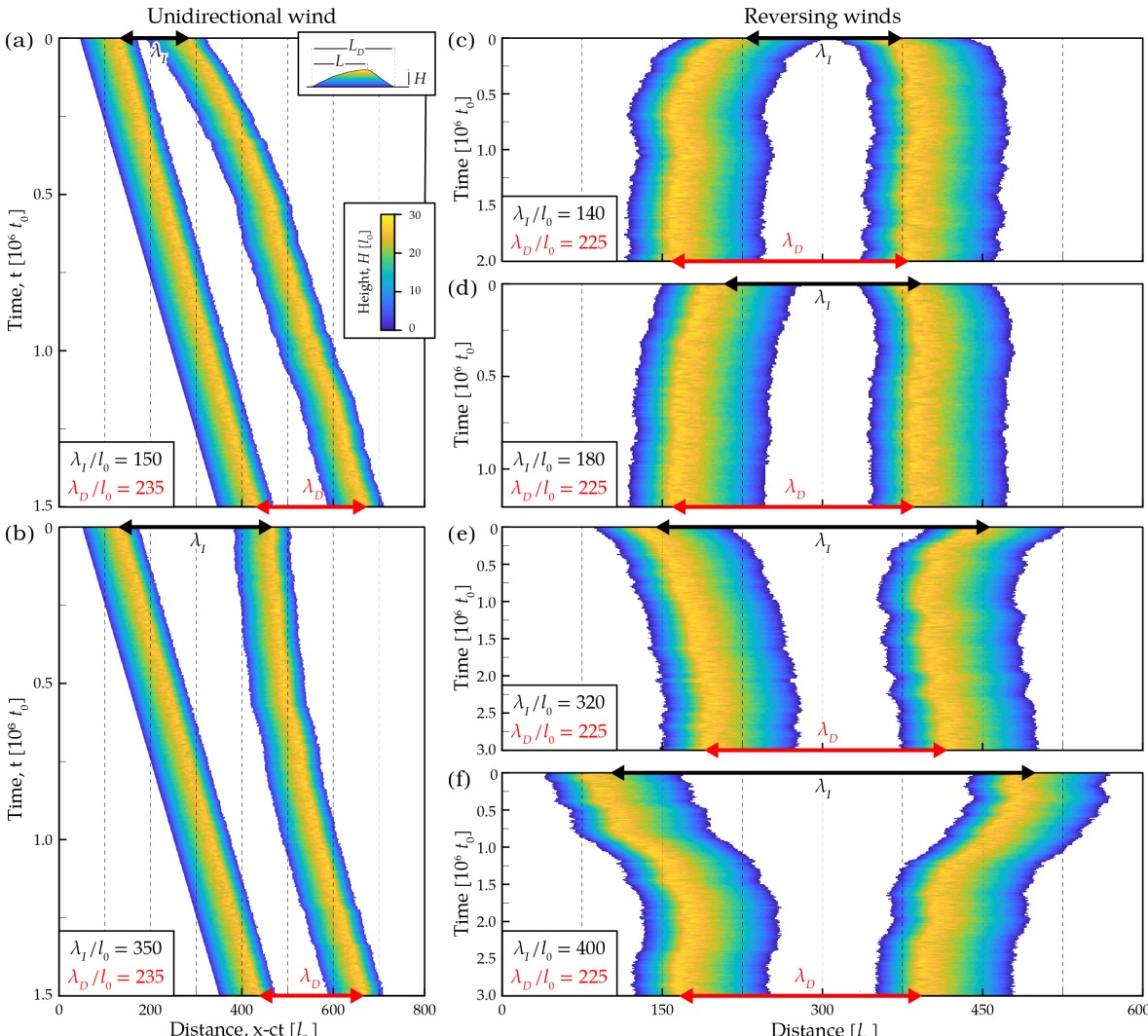

**Figure 2. Attraction and repulsion of isolated neighbouring dunes.** Space-time diagram of the dune elevation profile under **(a-b)** unidirectional and **(c-f)** symmetric reversing flows, using a period of flow reorientation $\Delta T = 10^4\, t_0$, a threshold shear stress $\tau_1/\tau_0 = 0$, and a dune size $S = 2000\, l_0^2$. Black and red arrows show the initial and steady-state spacings, $\lambda_I$ and $\lambda_D$, respectively. Isolated dunes attract ($\lambda_I > \lambda_D$) or repel ($\lambda_I < \lambda_D$) each other, eventually reaching a steady-state state characterised by an equilibrium distance and the same migration rates. In **(a-b)**, since the attraction and repulsion rates are two orders of magnitude lower than the migration rate under unidirectional flows, the position of the dune is shifted with respect to time by considering a constant speed $c = 7.7 \times 10^{-3}\, l_0/t_0$ for the visibility of the figure. Due to crest reversals, note the higher dispersion in the evolution of the dune spacing under reversing flows.

induced perturbations in the wake of dunes, a detailed analysis of the bed shear stress distribution in the model can be used to quantify these long-range interactions.

## 3.2 Control of bed shear stress distribution on attraction and repulsion

As shown by streamline curvatures and bed shear stress variations, the near surface flow over a single (Figs. 3a-b) and a pair of dunes (Figs. 3c-d) in the model presents similar variations. As illustrated in Figs. 3a-b for a single dune, the concave curvature at the dune toe produces a slight drop in shear stress, $\tau_{\text{s}}$, followed by streamline compression and an increase in $\tau_{\text{s}}$ on the stoss slope. Downstream of the crest, the streamlines diverge and a shear layer appears above a recirculation zone, which extends beyond the slip face and is characterised by a significant decrease in $\tau_{\text{s}}$. A few dune heights away from the crest, $\tau_{\text{s}}$ rises in the reattachment area, exceeding $\langle\tau_{\text{flat}}\rangle$. Then, $\tau_{\text{s}}$ decreases, becoming slightly inferior to $\langle\tau_{\text{flat}}\rangle$, before gradually converging towards the steady value, $\langle\tau_{\text{flat}}\rangle$. The results demonstrate an oscillatory behaviour of $\tau_{\text{s}}$-values in the wake of a dune, with a decrease in amplitude downstream. Considering a pair of dunes, while the integral of $\tau_{\text{s}}$-values on the stoss side of the upstream dune, $\langle\tau_{\text{s}}\rangle_{\text{D1}}$, is invariant, the integral of $\tau_{\text{s}}$-values on the stoss side of the downstream dune, $\langle\tau_{\text{s}}\rangle_{\text{D2}}$, will be subject to variations depending on its position along the oscillating wake generated by the upstream dune (Figs. 3c-d).

Difference in mean shear stress between the upstream and downstream dunes, $\Delta\langle\widehat{\tau_{\text{s}}}\rangle$, provides a means of quantifying long-range dune interactions by assessing the impact on transport rate of secondary airflow patterns in the wake of dunes (Figs. 3c-d). Figures 3e-g show that there is a linear dependence of the repulsion and attraction rates, i.e., variations in dune spacing, on the $\Delta\langle\widehat{\tau_{\text{s}}}\rangle$-value. Even if this linear dependence does not have the same slope for positive and negative values, the change in sign naturally explains the transition from repulsion to attraction, and the systematic convergence towards an equilibrium distance $\lambda_{\text{D}}$. At this equilibrium distance, the $\Delta\langle\widehat{\tau_{\text{s}}}\rangle$-value fluctuates around zero and, on average, the flow exerts a comparable influence on the overall transport rate on both dunes such that $\langle\tau_{\text{s}}\rangle_{\text{D1}} = \langle\tau_{\text{s}}\rangle_{\text{D2}}$. Given that dune morphology controls the structure of turbulent flow (Best and Kostaschuk, 2002; Lefebvre and Cisneros, 2023) and the subsequent distribution of shear stress over neighbouring dunes, it is necessary to investigate how dune shape may govern the equilibrium distance.

## 3.3 Dependence of the equilibrium distance on flow reversal frequency and dune shape

Figures 4a-b show how the dune aspect and shape ratios - measured before a flow reversal when the dune spacing has reached a steady state - depend on the period of flow reorientation $\Delta T$ (Rozier et al., 2019). In the limit of short periods ($\Delta T \to 0$), the distance travelled by the crest between two flow reversals tends to zero, and the dune has a triangular shape ($S/(HL_{\text{D}}) \to 0.5$) with slopes approaching the avalanche angle ($H/L \to 0.7$). For longer periods, the crest reversal distance increases, the dune has a gentler stoss slope, and a more rounded shape despite the systematic development of slip faces in the lee. When the period of flow reorientation becomes approximately 5 times longer than the dune turnover time, $T_{\text{D}}$, the dune migrates a significant distance after crest reversal, without changing shape until the next flow reversal. The steady aspect and shape ratio values are then determined by flow strength, as observed under unidirectional flows (Zhang et al., 2010).

These changes in dune shape modify the wake flow in such a way that the flow reversal frequency has also a direct impact on the equilibrium distance. Figure 4c shows that this equilibrium distance increases with the period of flow reorientation, converging towards the equilibrium distance observed for unidirectional flows. Nevertheless, if the period of flow reorientation

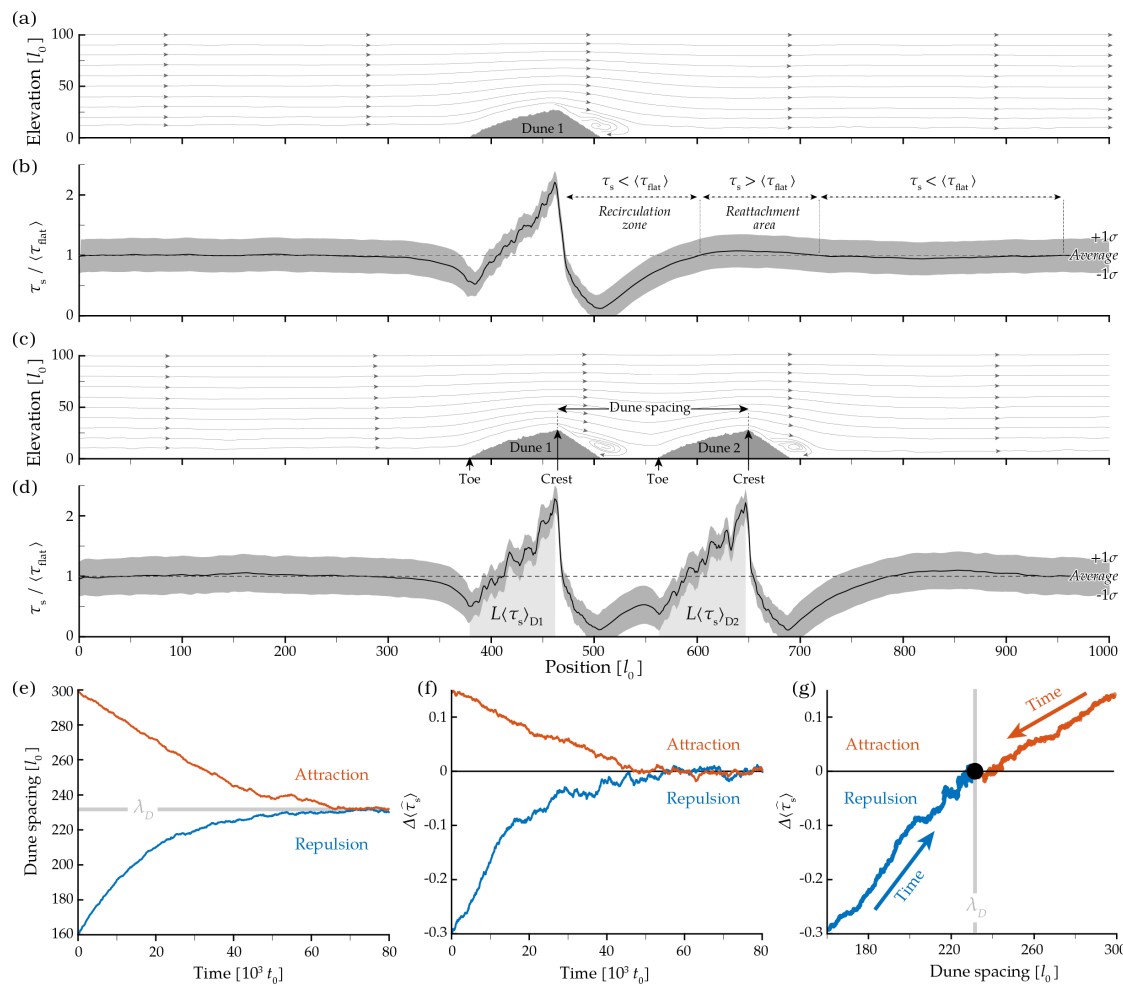

**Figure 3. Flow and bed shear stress over isolated neighbouring dunes.** Velocity streamlines and bed shear stress over **(a-b)** a single dune and **(c-d)** a pair of dunes with an initial spacing $\lambda_I = 180\,l_0$. Measurements of fluid flow are performed over static dunes of size $S = 2000\,l_0^2$ during $10^4\,t_0$. The black line and the dark grey area show the mean value and the dispersion at one standard deviation of the bed shear stress, respectively. These values are normalised by the average shear stress on a non-erodible bed measured away from any topography, $\langle\tau_{\text{flat}}\rangle$. **(a-b)** From the dune crest to the steady flow observed far downstream of the dune, an oscillatory behaviour of the bed shear stress with a decreasing amplitude is observed between zones where $\tau_s$ is successively inferior (i.e. recirculation zone), superior (i.e. reattachment area) then inferior to $\langle\tau_{\text{flat}}\rangle$. **(c-d)** The light grey areas are the integral of the bed shear stress measured between the toe and the crest of dunes, $L\langle\tau_s\rangle_D$ (see Methods). Note that $\langle\tau_s\rangle_{D1} < \langle\tau_s\rangle_{D2}$, indicating that dunes are in a repulsion state. **(e)** Evolution of dune spacing and **(f)** difference in mean shear stress, $\Delta\langle\hat{\tau}_s\rangle$ for a pair of dunes with an initial spacing smaller (blue) and longer (red) than the equilibrium distance, $\lambda_D$. **(g)** Relationship between dune spacing and difference in mean shear stress shows that this is a stable equilibrium.

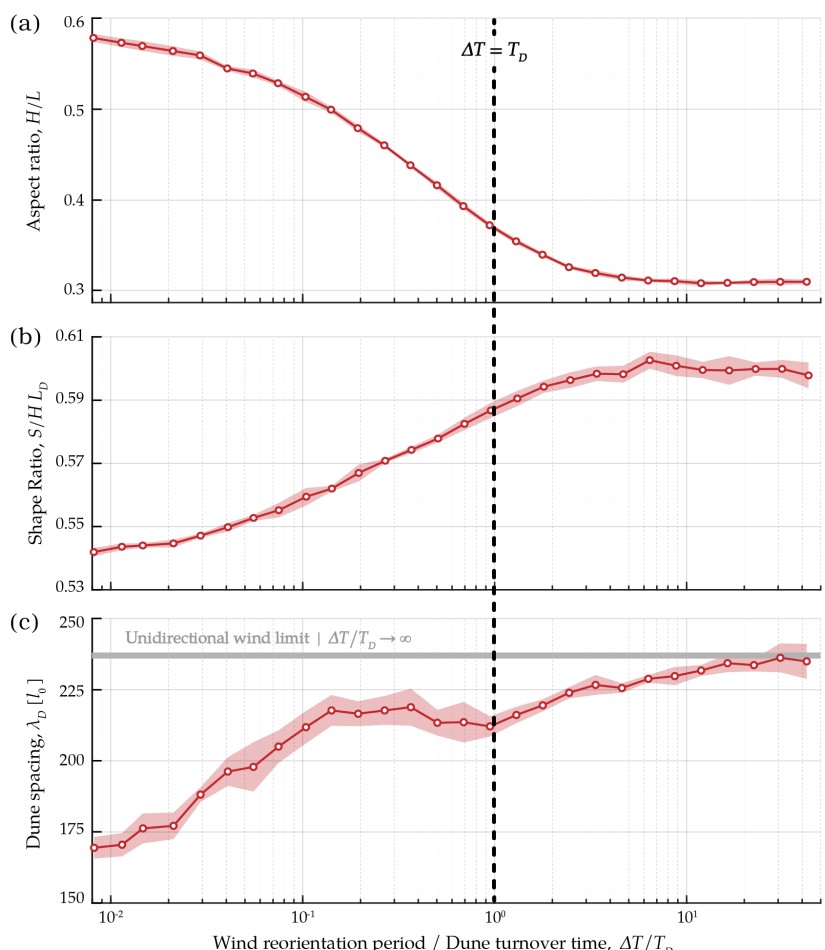

**Figure 4. Dependence of dune shape and spacing on the period of flow reorientation. (a)** Dune aspect ratio, $H/L$, **(b)** dune shape ratio, $S(H/L_{\mathrm{D}})$, and **(c)** equilibrium dune spacing, $\lambda_{\mathrm{D}}$, as a function of the period of flow reorientation, $\Delta T$, normalised by the dune turnover time, $T_{\mathrm{D}} = 1230\,t_0$ (i.e., time taken by the dune to travel over its own length). Measurements are performed after long time, when the steady-state has been reached ($t/\Delta T > 2 \times 10^3$), for a dune size $S = 1000\,l_0^2$, a threshold shear stress $\tau_1/\tau_0 = 0$, and an initial spacing $\lambda_{\mathrm{I}} = 160\,l_0$. The period of flow reorientation varies from 10 to $10^5\,t_0$ with regular intervals on a logarithmic scale. The shaded area shows the dispersion at one standard deviation using 10 simulations (see Methods).

is similar to the dune turnover time, the equilibrium distance drops noticeably, implying an additional contribution from the dynamics of crest reversal.

Crest reversals continuously modify the dune shape between an initial state and a final state selected by the frequency of flow reversals. For low $\Delta T/T_{\mathrm{D}}$-values, the crest reversal distance remains small, so dunes maintain high aspect ratio and low shape ratio between two flow reversals (Fig. 5a, $\Delta T/T_{\mathrm{D}} = 0.07$). For high $\Delta T/T_{\mathrm{D}}$-values, the crest reversal phase is negli-

gible compared to the migration phase, during which dune shape is characterised by a low aspect ratio and a high shape ratio (Fig. 5a, $\Delta T/T_D = 41$). For intermediate $\Delta T/T_D$-values, close to 1, the crest undergoes a complete reversal lasting almost the entire period between two flow reversals. The aspect ratio decreases as the stoss slope becomes gentler. The shape ratio decreases and then increases as the crest migrate over the centre of mass of the dune (Fig. 5a, $\Delta T/T_D = 1.1$).

Between flow reversals, Figure 5b shows that the difference, $\Delta\langle\widehat{\tau}_s\rangle$, in mean shear stress between the two dunes evolves not only with respect to their spacing, but also according to their shape during the crest reversal. After a flow reversal, the potential equilibrium distance at which $\Delta\langle\widehat{\tau}_s\rangle = 0$ first decreases, then increases as the crest reversal is complete and the dune begins to migrate at constant shape (Figs. 5b-c). Therefore, according to the primary impact of the dune aspect ratio, the prevailing dune shape between flow reversal weights the steady-state equilibrium distance between repulsion and attraction. This modulation

associated with the crest reversal dynamics naturally explains the apparent reduction in the equilibrium distance for values of $\Delta T/T_D$ approaching 1 (Fig. 4c). Indeed, for these specific flow periods, the more triangular transient dune shapes during crest reversals reduce the difference in bed shear stress between neighbouring dunes. Consistent with the repulsion and attraction rates observed in Figure 3g, Figure 5b shows that repulsion rate is twice as larger as the attraction rate because the magnitude of the flow-induced perturbation, expressed by $\Delta\langle\widehat{\tau}_s\rangle$, decreases with distance with an oscillatory behaviour. This oscillatory

behaviour of $\Delta\langle\widehat{\tau}_s\rangle$-values also correlates with the variations of basal shear stress observed in the wake of a dune (Fig. 3b), explaining repulsive and attractive regimes. The characteristic wavelength of this behaviour is given by the equilibrium distance, and increases as the dune aspect ratio decreases.

## 3.4    Dependence of the equilibrium distance on flow strength and dune size

Figure 6 shows how the equilibrium distance between two equal-sized dunes varies as a function of the dune size and transport

threshold (i.e., flow strength) for a range of flow reorientation periods. Under reversing flow regimes, the equilibrium distance increases when the dune aspect ratio decreases, i.e., when the transport threshold and/or the period of flow reorientation increase. For long periods, it eventually converges to the equilibrium distance observed under unidirectional flows. The drop observed for intermediate aspect ratio values ($0.35 < H/L < 0.45$) reflects the continuous change in dune shape during a period of constant flow orientation, i.e., when $\Delta T/T_D \approx 1$ (greenish colors in Fig. 5). Under unidirectional and reversing

flow regimes, the equilibrium distance increases with dune size, while the aspect ratio of the dunes remains constant. This dependence of the equilibrium distance on dune size is greater when the transport threshold is higher (i.e., when the flow strength is lower). All these numerical results are obtained for dunes with aspect ratios that range from 0.6 (i.e., triangular dunes) to 0.2 (i.e., asymmetric dunes with slip faces). These values are consistent with the aspect ratios of dunes formed under bidirectional flows on Earth (Lancaster, 1988; Bristow et al., 2000).

It also should be noted that the reversing dune shapes for high $\Delta T/T_D$ are flatter than those observed under unidirectional flows (for same dune size and transport threshold), underlining the long-term morphological impact of crest reversal on dune shapes (Fig. 5). This difference in dune shapes is largely determined by the transport processes that occur after flow reversals on the gentle slope of the new lee side.

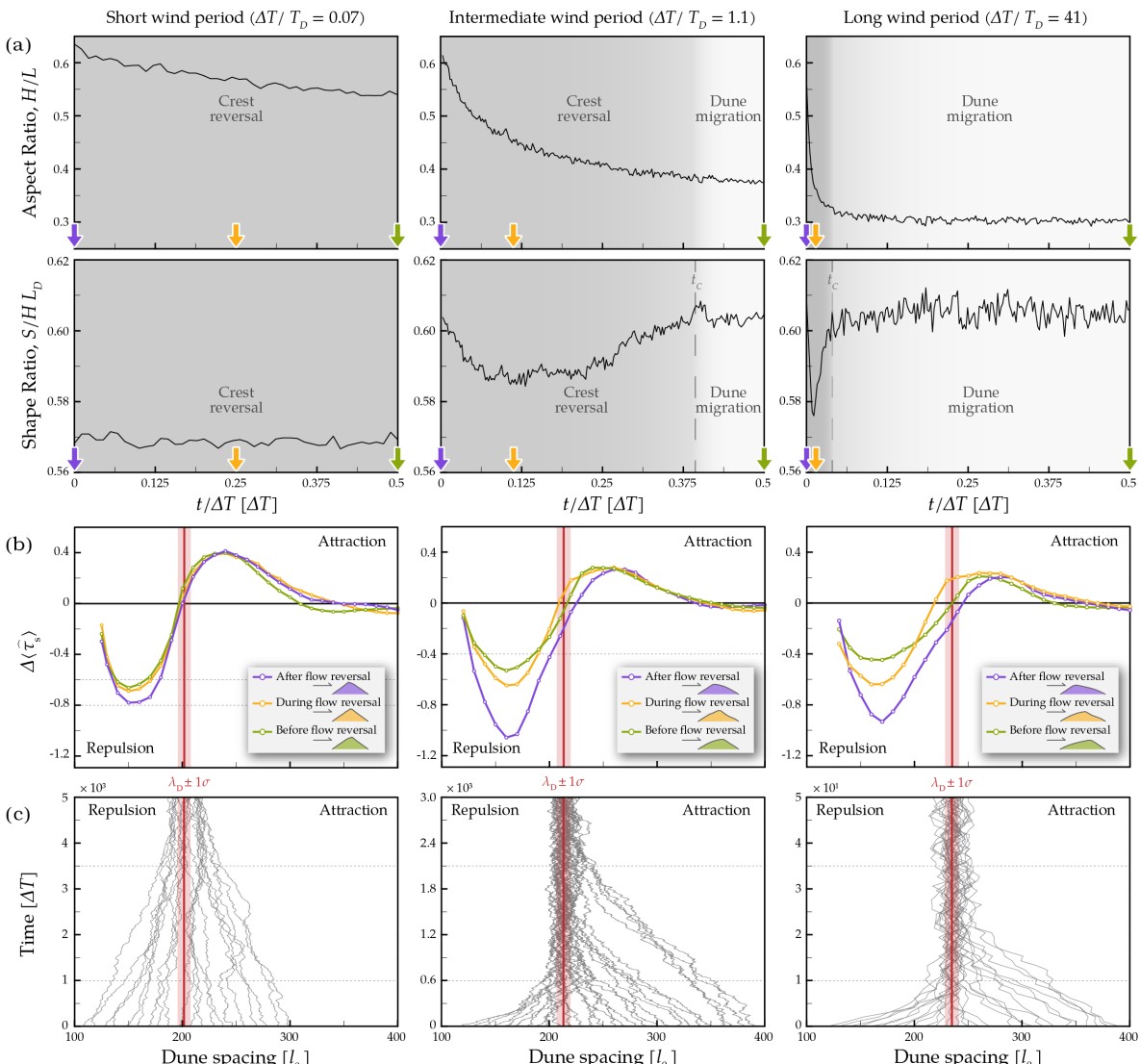

**Figure 5. Control of crest reversal dynamics on the equilibrium distance. (a)** Evolution of dune aspect ratio (top) and shape ratio (bottom) between flow reversals, for short (left), intermediate (middle) and long periods of flow reorientation (right). The vertical dotted line shows the crest turnover time, when the slip face has fully reversed. Purple and green arrows show two times just after, and just before flow reversals, the yellow arrow a time during crest reversal. **(b)** Difference in mean shear stress between dunes $\Delta\langle\widehat{\tau}_s\rangle$ as a function of interdune spacing at 3 different times during a period of constant flow (see arrows in **(a)** for this times and the corresponding dune shape). **(c)** Evolution of the dune spacing for different initial distances, $\lambda_I$ (grey lines). Red line and shaded area show the mean value $\lambda_D$ of the steady-state spacing over long times and the dispersion at one standard deviation. This equilibrium distance is selected by the prevailing dune shape during a flow cycle.

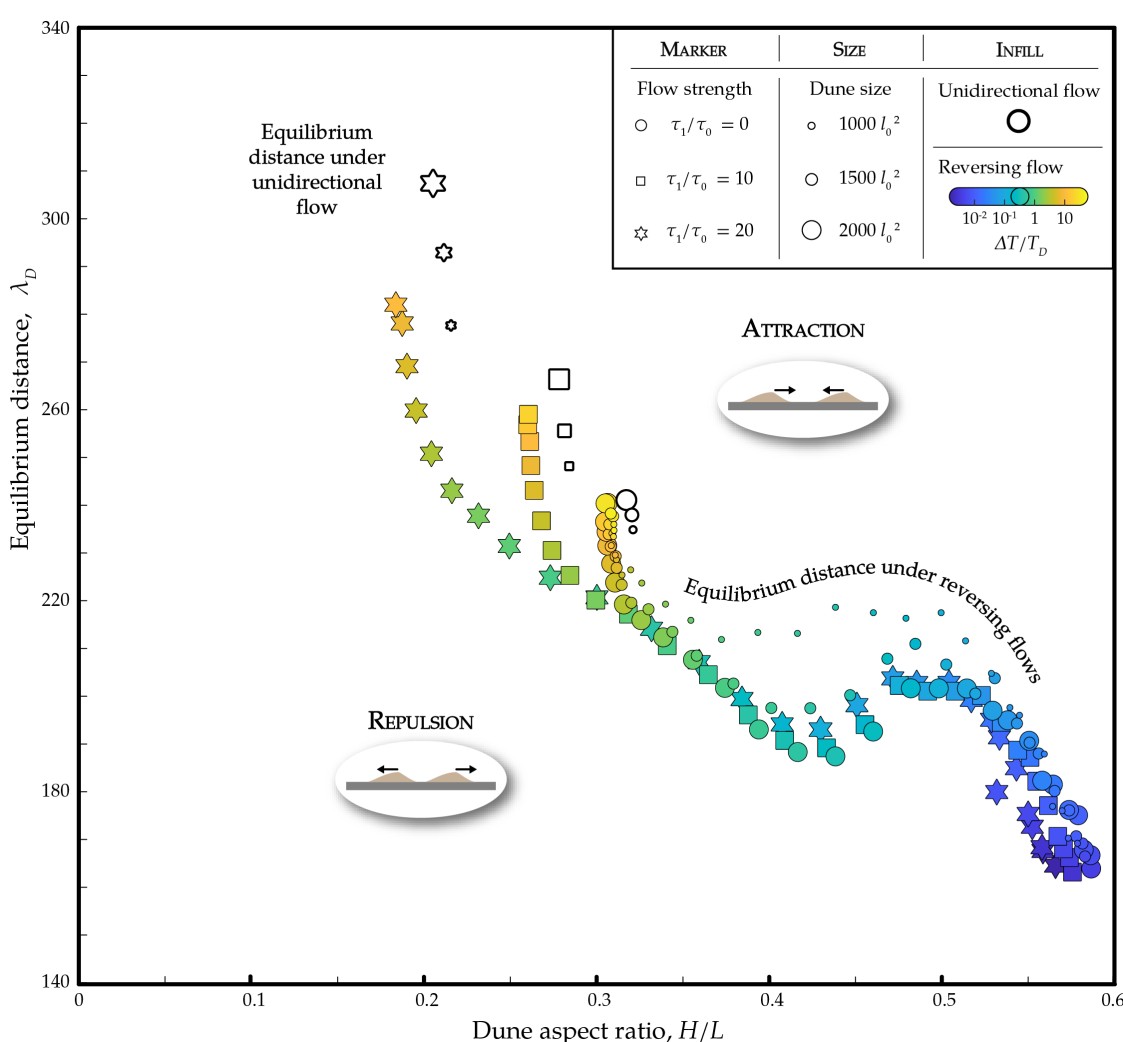

**Figure 6. Dependence of the equilibrium distance on flow conditions and dune size.** Steady-state dune spacing with respect to dune aspect ratio for a pair of dunes under unidirectional (white symbols) and reversing flow regimes (coloured symbols) for different dune sizes, flow strengths and periods of flow reorientation (see inset). Measurements of dune spacing and aspect ratio are performed after long time, when the steady-state has been reached ($t/\Delta T > 2 \times 10^3$), and before flow reversals. The equilibrium distance increases with decreasing dune aspect ratio, i.e., with increasing transport threshold and period of flow reorientation. For long periods, the equilibrium distance converges to the one observed under unidirectional flows. For intermediate periods close to the dune turnover time, i.e., $\Delta T/T_D \approx 1$ (greenish colour), the continuous change in dune shape between two flow reversals explains the different regimes observed at intermediate dune aspect ratio ($0.35 < H/L < 0.45$).

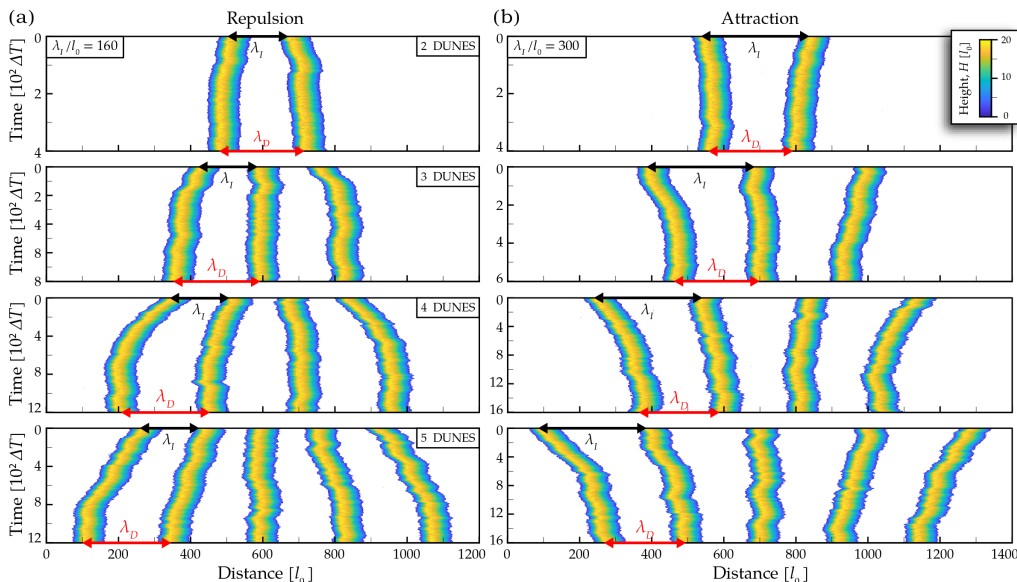

**Figure 7. Emergence of a characteristic wavelength from long-range interactions in dune fields. (a)** Repulsion and **(b)** attraction of isolated neighbouring dunes using $\Delta T = 5 \times 10^3\ t_0$, $\tau_1/\tau_0 = 0$, and $S = 1000\ l_0^2$. From the top to bottom, we consider 2, 3, 4 and 5 dunes. Black and red arrows show the initial and steady-state spacing, $\lambda_I$ and $\lambda_D$, respectively.

## 3.5 Equilibrium distance and dune wavelength

Wake flow perturbations provide an elementary mechanism for the development of a characteristic wavelength in dune fields with non-erodible interdune areas submitted to bidirectional flows. Using reversing flows, Figure 7 shows that attraction and repulsion mechanisms contribute to the dynamics of a population of isolated dunes in the model, resulting in a constant spacing that is similar in magnitude to the equilibrium distance observed between two dunes. As a consequence, a periodic steady-state dune pattern emerges and a mean wavelength can be measured. However, when dealing with more than two dunes, the flow and the subsequent shear stress distribution become more variable. It is challenging to directly extrapolate the regimes observed for two dunes because the mean shear stress on the successive dune slopes does not show systematic variations along the flow direction.

## 4 Discussion

While a dynamical system can predict the impact of long-range repulsion on the dynamics of neighbouring dunes observed in subaqueous laboratory experiments (Bacik et al., 2020, 2021a, b), we numerically show here that we can quantify the impact of these long-range interactions on the dynamics of a pair of dunes in reversing flow regimes. Under these conditions, subtle variations can be identified according to the relative migration rate between the dunes. Nevertheless, they are not self-similar

in shape over time, and for different flow periods or flow strengths. The mechanism of repulsion is then primarily influenced by dune shape, which modifies the spatial structure of the turbulent wake and the resulting distribution of the bed shear stress downstream.

## 4.1 An oscillatory behaviour of the difference in mean shear stress between two neighbouring dunes

Experimental measurements of the turbulent flow (Bennett and Best, 1995; Frank and Kocurek, 1996; Walker and Nickling, 2002; Dong et al., 2008; Palmer et al., 2012; Bristow et al., 2019, 2021; Cai et al., 2021) and computational fluid dynamics methods over consolidated dunes (Stoesser et al., 2008; Omidyeganeh et al., 2013; Anderson and Chamecki, 2014; Smith et al., 2017; Wang et al., 2017; Wang and Anderson, 2018; Wang et al., 2019; Jackson et al., 2020; Love et al., 2022) have demonstrated that the wake-induced turbulence perturbs the bed shear stress downstream with a magnitude that decreases with distance. To determine the impact of these long-range interactions on dune morphodynamics, the associated changes in transport rate need to be quantified. Our discrete numerical approach, which couples models of sediment transport and fluid flow, allows to capture this feedback mechanism and examine its dependence on flow strength, as well as on dune shape and size. At this stage, it appears that attraction and repulsion rates can be estimated by comparing the mean shear stresses on the stoss slope of isolated neighbouring dunes. The highest rates are associated with repulsion when dunes are in close proximity to each other, while at greater distances, the maximum attraction rates are almost twice as low (Figs. 3e-g), supporting the decay of the wake flow perturbation of the bed shear stress with distance. It is now necessary to link these differences in dune migration rate and mean shear stress with unsteady turbulent flow structures.

Compared with computational fluid dynamic models that solve the Navier-Stokes equations, our discrete lattice gas model for fluid flow may not have sufficient accuracy to evaluate in greater detail the turbulence intensity in the various flow regions observed in the wake of dunes. It is possible to reproduce the recirculation zone with a length of a few dune heights, but it is difficult to identify secondary flow regions in the reattachment area. These are not critical for our conclusions, which can now be tested in laboratory experiments and advanced numerical methods by quantifying the mean bed shear stress on dunes according to the attraction and repulsion states.

As a result of long-range flow perturbations, there is a transition from repulsion to attraction with increasing dune spacing in the numerical model. It can be regarded as a stable equilibrium distance resulting from the oscillatory behaviour of the flow in the wake of a dune (Fig. 3b), and the subsequent difference in mean shear stress exerted on the slope of two neighbouring dunes (Fig. 5b). According to this oscillatory behaviour, an unstable equilibrium should exist at a greater distance, but the decreasing amplitude of the signal makes it impossible to evaluate with our model. Another unstable equilibrium at shorter distance can be measured by the difference in mean shear stress, but as it corresponds to a short interdune distance, it often leads to collision under reversing flows.

The real question is the origin of the oscillation that causes the transition from repulsion to attraction. In the model, the signal combines the flow disturbance caused by the upstream dune and its response to the topography of the downstream dune. This could be particularly significant when the reattachment point of the upstream recirculation zone overlap with the drop in shear stress commonly observed at dune toe (Fig. 3b). Independently, the flow separation over the lee side of dunes may gives

birth to periodic wake structures (Zheng et al., 2019). In this case, the flow confinement and the range of Reynolds numbers accessible to the model may have an impact on the equilibrium distance between dunes, considering the dynamic interactions between the wake flow and the topography of the downstream dune. As shown in Gao et al. (2015b) for the periodicity of giant dunes, the depth of the flow is likely to exert a positive dependence on the equilibrium distance, maintaining the repulsion/attraction mechanism proposed here. Further investigations need to be done in order to quantify this dependence. However, if the transition to the state of attraction occurs at greater distances, the intensity of repulsion also weakens with increasing distance (Fig. 5). The equilibrium distance would then no longer correspond to a stable equilibrium, but rather to a maximum distance above which the perturbation of the flow becomes negligible, and dunes no longer interact.

## 4.2   The impact of crest reversals on the equilibrium distance

Based on numerical simulations, the equilibrium distance can be accurately estimated when a pair of equal-sized dunes converges over long times to a constant spacing under reversing flow conditions. This equilibrium distance increases with decreasing flow strength and increasing dune size (Fig. 6), as the dune aspect ratio decreases (Zhang et al., 2010; Gao et al., 2015a). Flow reversals particularly highlights the dependence of the equilibrium distance on the dune aspect ratio. Indeed, the period of flow reorientation determines dune shape (Rozier et al., 2019). Frequent flow reversals produce steeper and symmetrical dunes, while less frequent reversals lead to the formation of asymmetric dunes with a gentle stoss slope and an avalanche face in the lee (Fig. 5). In addition, crest reversals are associated with a sudden increase in the apparent dune aspect ratio, and nonlinear variations in sand flux on the dune flanks (Gao et al., 2021). These changes in shape significantly modify the bed shear stress distribution on dunes, and the attraction and repulsion states during crest reversals. As a result, the equilibrium distance increases with the period of flow reorientation, except when this period is of the same order of magnitude as the dune turnover time. In this case, the change in dune shape is complete, but the dune has not yet migrated a significant distance. When the period of flow reorientation is much longer than the dune turnover time, the effect of crest reversals becomes negligible, and the equilibrium distance becomes similar to that observed under unidirectional flow regimes. However, numerical results indicate that reversing flow regimes increase the stoss-side length, offering an explanation for the width of the largest dunes submitted to multi-directional winds in nature (Fig. 6).

## 4.3   An alternative mechanism for dune wavelength selection in areas of limited sand supply

Transition and repulsion states can be extrapolated to all wind regimes and natural environments with limited sand supply, potentially influencing dune migration rate and the wavelength observed in linear dune fields. Under unidirectional wind regime, the organisation of barchan dune fields essentially relies on calving and collision processes due to the negative dependence of migration rates to dune size (Hersen et al., 2002; Durán et al., 2009; Génois et al., 2013; Worman et al., 2013; Robson and Baas, 2024). Wake flow perturbations could act as a second order component of the dynamics of barchan dune fields, locally modifying the migration rate of individual barchans located in the wake of one another at local scale. For example, before collision, an impacted dune should reduce (attraction) and then increase its migration rate (repulsion) as a smaller impacting dune is approaching (Fig. 8a). This effect could be captured in barchan dune fields where collisions occur (Katsuki et al.,

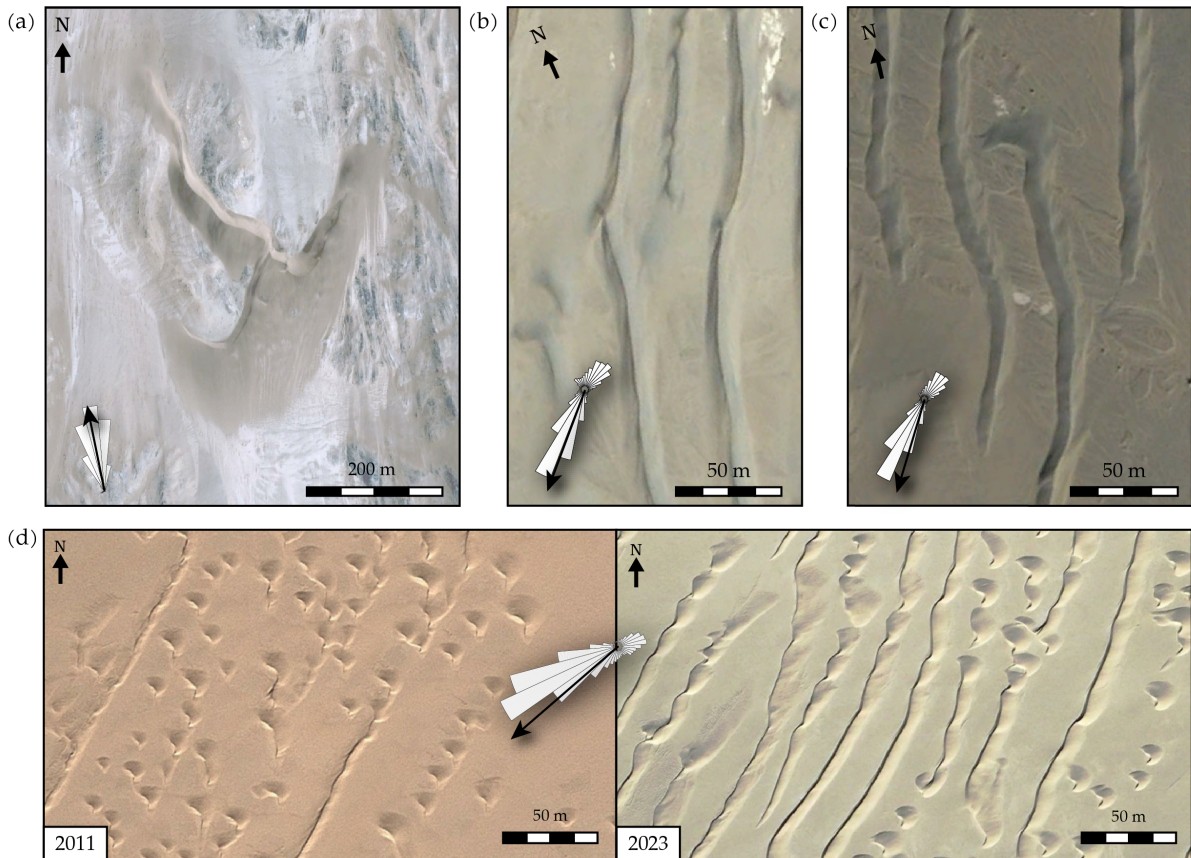

**Figure 8. Long-range dune interactions in terrestrial dune fields. (a)** Collision of barchan dunes south of the Namib Desert, Namibia (26.85°S, 15.30°E). **(b-c)** Interacting elongating dunes at the southern edge of the Taklamakan Desert, China (**b** 37.43°N, 83.79°E, **c** 37.53°N, 84.02°E). **(d)** Emergence of periodic linear dunes (2023) from a population of barchans (2011), Taklamakan Desert, China (37.66°N, 82.61°E). Sand flux roses and resultant drift directions (black arrows) are calculated from wind data provided by the ECMWF ERA5-Land reanalysis(Hersbach et al., 2019) from January 1, 2010 to December 31, 2020. Image credits: © Google Earth.

2005; Hersen and Douady, 2005), and also as defects propagate in transverse dune fields (Gao et al., 2015b). In both cases, the dependence of these flow-induced interactions on the dune aspect ratio could be detected by taking advantage of the changes in shape of the different longitudinal dune slices involved. However, the distance ranges over which the states of attraction could be felt are likely to be highly dependent on the specific dune configuration, considering 3D flow channelling effects, transverse sand fluxes or permanent exchanges of mass related to free flux between dunes.

Under multi-directional wind regimes, the transition from repulsion to attraction with increasing dune spacing is critical (Fig. 6). First, because under these conditions, and for similar wind strengths, the resultant migration rates of dunes are lower. Differential migration rates caused by wake-flow perturbations - which we show represents between 1% to 3% of the dune

migration rate under unidirectional flow - can therefore become a dominant contribution of dune field dynamics. Second, because it is associated with an equilibrium distance that provides a characteristic wavelength for linear dune fields (Figs. 7-8), whatever the underlying growth mechanism. For example, it could explain why periodic linear dunes can spontaneously form from a field of isolated dome or barchan dunes (Fig. 8d).

In nature, most linear dune fields form under bidirectional winds with a divergence angle smaller than 180° and transport ratio different than 1. Therefore, depending on the wind orientation, the apparent dune aspect ratio and interdune distance vary, modifying the steady-state spacing and the attraction/repulsion state of dunes. Then, compared to the optimal 2D conditions and the symmetric reversing winds used here, natural wind regimes introduce an additional level of complexity into the long-range flow-induced interactions between dunes, and the potential selection of linear dune wavelength. Further investigation is required to ascertain whether this wavelength is affected by the dominant wind (longer distance), the minor wind (shorter distance), or a combination of both.

Flow-induced interactions are also systematically expressed in dune fields when the interdune distances readjust according to the formation, migration and elongation of new dunes (Figs. 8b-c). As individual dunes grow, these long-range interactions continuously adapt the spacing between dunes to their size and length. This spacing-selection mechanism is likely to depend on the wind regime (Lü et al., 2018; Rozier et al., 2019; Ma et al., 2022), especially on the period of flow reorientation, through its continuous effect on dune shape. On Earth, where seasonal winds sets a typical period of one year, the existence of an equilibrium distance is clearly visible for small dunes that are relatively close to each other (Fig. 8). However, the results of the numerical model cannot yet predict the aspect ratio and wavelength observed within giant dune fields (Andreotti et al., 2009; Gunn et al., 2022). As they integrate atmospheric conditions over long time, the wind regime, the flow depth and the sediment supply are continuously changing when compared to the time scale of dune patterns. Thus, fields of giant dunes are likely to remain in an out-of-equilibrium state (Gunn, 2023). In addition, other processes related to collision, coarsening or superimposed bedforms are expected to influence the selection of the interdune distance.

## 5  Conclusions

Using a numerical model in 2D, we demonstrate that neighbouring dunes in areas of limited sand supply can interact via the perturbation of the flow induced by their topography. According to these long-range interactions induced by the flow, there is an equilibrium distance below and above which repulsion and attraction mechanisms occur. This study is an incremental step towards a complete classification of long-range dune interactions in nature, based on a simplified numerical setup where exchange of mass can be removed. In addition to field observations and keeping in mind that the 3D numerical setups have a huge computational cost, future research could implement advanced computing methods for turbulence and sediment transport in order to go further in quantifying these flow-induced mechanisms. As for the intricate relationship between shear stress and dune shape, one may hope to identify specific flow properties on neighbouring dunes separated by a non-erodible bed. In order to complement the close relationship between shear stress and dune shape, these studies will also allow for the introduction of the contribution of long-range interactions between neighbouring dunes in zones of low sand availability.

*Code availability.* The installation package of the ReSCAL dune numerical model is available in the following link. This model is submitted to a GNU General Public License - Copyright (C) 2011 (https://www.ipgp.fr/ rozier/rescal/rescal.html)

*Data availability.* All data and materials used in this paper are available under request, which should be addressed to J.V. and C.N.

*Author contributions.* J.V., C.N., and L.B. designed the study. J.V. carried out all statistical data analysis and numerical simulations. O.R. adapted the numerical model. S.C.P. conducted the laboratory experiments. J.V. and C.N. wrote the manuscript with contributions from all authors.

*Competing interests.* The authors declare that they have no conflict of interest.

*Acknowledgements.* We acknowledge financial support from Laboratoire d'Excellence UnivEarthS Grant ANR-10-LABX-0023, Initiative d'Excellence Université Paris Cité Grant ANR-18-IDEX-0001 (AEOLAND project), and the EOLE project of the French National Research Agency Grant ANR-23-CE56-0008. We thank Pascal Hersen for the images of the subaqueous dune experiments shown in Figure 1a. We also thank two anonymous reviewers, the handling editor Andreas Baas, and Yuanwei Lin for their insightful comments.

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
