# Peer review of "Equilibrium distance from long-range dune interactions"

_EGUsphere, 2024_

## Author Comment (AC1)

REVIEW - EGUSPHERE-2024-1634
**Equilibrium distance from long-range dune interactions**
Earth Surface Dynamics

**Reviewer #1**

**General comments:**

☑ *The choice of a 1:1 aspect ratio of the sand cells in ReSCAL*

To the knowledge of this reviewer, the effect of changing the elementary aspect ratio (i.e. height to length or width) of the sand slabs (cells) from which the bedforms are formed has never been investigated (other than a brief mention in Appendix A of *Narteau et al. (2009)*). Thus, one cannot be certain whether the results presented here are universal outputs of the model or hold only in the specific case of a 1:1 cell height:width ratio.

Changing the elementary aspect ratio may affect coupling between the bedform and lattice gas since the bedforms are likely to be comprised of a greater number of smaller sand slabs and so the removal of a single slab due to erosion/transport will have less of an impact on the shape/size of the bedform and thus result in a smaller perturbation of the flow. Changing the height of the slabs would also affect the avalanche procedure which could in turn influence the shape (and migration etc) of the bedforms.

In their review, Reviewer #1 highlights the significance of the choice to utilize a square grid with a 1:1 cell length:height ratio (*l/h*) in our ReSCAL dune model. The reviewer queries the impact of this choice on the model properties, the avalanche angle, the overall dune size and shape, the fluid flow, and therefore on the dynamical and flow-induced mechanism selecting the dune equilibrium distance, which has been demonstrated to depend on dune shape. All related specific comments are listed below:

→ *Influence on model properties*

- Line 80 - "*The physical environment is fully described by a regular lattice of square (2D) or cubic (3D) cells with an elementary length scale $l_0$*". This sentence makes it clear that this work considers only the case that the height and length of the cells are equal (i.e. a square lattice) and changing this aspect ratio (e.g. a rectangular lattice) has not been considered.
- Line 114 - "*The elementary length and time scales, {$l_0$, $t_0$}, and the threshold shear stress τ1 are entirely defined with respect to the dune instability using the most unstable wavelength, the mean flow strength and the corresponding saturated sand flux*". In *Narteau et al. (2009)*, the height:length ratio of the cells in all simulations shown was set to unity and it is not clear in that work how changing the ratio would impact the results due to both flow-coupling changes and avalanche changes as previously mentioned.
- Line 121 - "*Theoretically, this rescaling strategy also permits the model to overcome the fundamental limitation related to the arbitrary choice of the elementary length scale inherent to a cellular automaton approach*" – Same comment

→ *Influence on avalanche angle*

- Line 85 - "*To take into account avalanche processes, we impose a repose angle of 35° for the granular material*". How is this angle of repose achieved, surely at an angle of 35 degrees two adjacent cells that differ by a single slab in height would avalanche and so the only stable state would be a flat bed. If two adjacent cells are able to have a difference of a single slab (which is what this reviewer believes was

applied in e.g. *Zhang et al. JGR (2010)*) then the angle of repose is in fact 45 degrees, which is higher than stated here and what it should be for aeolian bedforms.

→ *Influence on dune size and shape*

- Line 136 - "*We save the topography built by the immobile sedimentary cells at the end of each flow period to measure the length $L_D$ and height $H$ of the dunes, as well as the length of their stoss slope $L$ and the interdune spacing $\lambda_D$ (i.e., distance between the two centres of mass)*"
  Again, these morphological characteristics might be influenced by choice of the cell shape.
- Line 172 - "*In the reattachment area, a few dune heights away from the crest, the bed shear stress increases and becomes higher than the steady value observed on a flat bed, before converging to this steady value*". The bedforms shown in this work are around 30 cells high, how would this "a few dune heights" change if the cells in the model were e.g. 1/10 as high as they are long and so dunes were 300 cells high?
- Section 3.3 – This section goes into detail about the reshaping timescale of the dunes and their shape. All of these processes are likely to be affected by the shape of the cells. If it turns out that they are unaffected then that is an interesting result worth including.
- Line 219 - "*The drop observed for intermediate aspect ratio values (0.35<H/L<0.45) reflects the continuous change in dune shape during a period of constant flow orientation*"
  Again, how do the cells affect these aspect ratios and dune size?
- Line 238 - "*The mechanism of repulsion is then primarily influenced by dune shape, which modifies the spatial structure of the turbulent wake and the resulting distribution of the bed shear stress downstream*". Given this, it is even more important to understand how the cell shapes are influencing the bedform shape.

→ *Influence on fluid flow*

- Line 164 - "*Since this equilibrium distance appears to be controlled by flow-induced perturbations in the wake of dunes, a detailed analysis of the bed shear stress distribution in the model can be used to quantify these long-range interactions.*" In Appendix A of Narteau et al. (2009) it is conceded that changing the aspect ratio of the cells in the model would "allow for the modeling of smaller structures of the flow". Thus the results here are certain to be influenced by the choice of a square lattice.
- Line 262 - "*This could be particularly significant when the reattachment point of the upstream recirculation zone overlap with the drop in shear stress commonly observed at dune toe*". As mentioned, how is the distance of the reattachment changed by changing the cell height in this model?
- Line 280 - "*These changes in shape significantly modify the bed shear stress distribution on dunes, and the attraction and repulsion states during crest reversals.*" - Same comment as previously

The shape of dunes is the result of the dynamic interactions between topography, wind flow and sediment transport. Consequently, the shape of dunes should not depend on the numerical method used. In discrete models, such as cellular automata models, the aspect ratio of the cells should not be a parameter of interest and should not impact dune morphodynamics.

All the confusion stems from the first generation of cellular automaton dune models developed at the end of the last century (*Werner, 1995; Werner & Kocurek, 1997, 1999*), for which the aspect ratio of the elementary cells, the so-called slab, was an intrinsic parameter of the model allowing the aspect ratios of dunes to be set arbitrarily. As this first generation of dune model is based on a transport length expressed in units of slab and does not take into account the impact of dune topography on basal shear stress and sediment transport, the horizontal dilation of elementary cells proved an expedient method for representing realistic stoss slopes. This issue was addressed by *Momiji et al. (2000)* and partially resolved by modifying

the local transport length to account for the acceleration of wind speed on the dunes. Nevertheless, the cell aspect ratio was once again set at *l/h = 3* in accordance with the previously established arbitrary parameter.

In the ReSCAL dune model (*Narteau et al., 2009),* which is used in this study, this issue was fully solved as the entire model is based on the dynamic interaction between topography, basal shear stress and sediment transport using the couplings between two cellular automata of sediment transport and turbulent flow. The combination of these two cellular automata induces several constraints which are discussed hereafter.

In the lattice gas cellular automaton, the collision rules ensure the conservation of both mass and momentum, necessitating that the norm of velocity vectors of fluid particles be identical in horizontal and vertical directions. Thus, the displacements of fluid particles have to be implemented on a regular lattice of elementary square cells with *l/h = 1*. In order to fit with the fluid flow model, the cell aspect ratio of the sediment transport model is set at *l/h = 1*. In the ReSCAL dune model, the aspect ratio of the elementary cell is therefore not a free parameter and should not be treated as such.

However, as described in *Narteau et al. (2009)* and thanks to the modular structure of ReSCAL, we can obviously change the aspect ratio of the elementary cells and adapt the resolution of the square lattice of the lattice-gas model to fit perfectly into the theoretical rectangular lattice of the sediment transport model with *l/h = {2, 3,…, n}*. This will lead to several implications of the model, which are discussed below.

→ *Avalanche slope*. Whatever the aspect ratio of the elementary cells, slopes of 90° exist as soon as we consider a cubic or rectangular geometry. This is not an issue as it is clear that a microscopic representation of cellular automaton approaches is not an exact representation of the real world. The realism of these discret models relies on patterns of interactions averaged over a large number of cells. The use of a square (*l/h = 1*) rather than rectangular elementary cells (e.g. *l/h = 3*) determines the threshold angle formed by a step between two neighboring piles of sediments, which are respectively 45° and 18°. Considering a threshold angle for avalanche at 35° (as defined for dry sand in nature), it appears as a wrong assumption to consider an elementary square grid since the threshold angle is necessarily superior to the avalanche angle. However, in our model, the slope is averaged locally on five neighboring piles of sediments, blocking avalanche transition as long as the local slope is inferior to the threshold avalanche angle. This process is controlled by an avalanche module in ReSCAL based on a diffusion with threshold mechanism to improve the sedimentary patterns associated with gravity-driven granular flows, forming dune slip faces with a slope of 35° rather than 45° (find more details in *Zhang et al., 2014* and in the Supplementary Material S1 of *Gao et al., 2016,* where the same procedure also reproduce segregation and stratification patterns observed on avalanche slopes of bidisperse granular materials). We agree with Reviewer #1 that the manuscript would benefit from additional clarifications regarding the avalanche slope. This has been addressed in the revised version of the paper (see Section 2.1.1).

→ *Dune shape*. Considering that the avalanche angle is determined over few elementary cells and does not depend on the cell aspect ratio, the slope of the lee face would remain unaltered with rectangular elementary cells, although it would favor a higher vertical resolution of dune topography. Given the shape of dune in ReSCAL relies on the physical interactions between topography, basal shear stress and sediment transport (i.e. coupling between two models), the slope of the stoss face should not be dependent on the aspect ratio of the elementary cells

→ *Turbulent flow structures*. Reducing the height of cells and using a rectangular grid would also allow the formation of smaller and more detailed flow structures. However, it would have no impact on the overall distribution of the basal shear stress on dune faces. Indeed, the basal shear stress is roughly estimated from the longitudinal velocity of the flow along the normal vector to the topography at a distance of 10 $l_0$, allowing to overcome the variability in flow structure resolutions depending on cell

shapes close to the bed. Given the basal shear stress distribution is the main factor controlling the differential dune migration rate induced by the wake flow perturbation, the mechanism we propose in this paper would remain unchanged whatever the cell shape, but may require to change the distance over which the shear stress is computed.

Despite these implications that do not fundamentally modify the model and the mechanism proposed, choosing a rectangular elementary cell in the model of sediment transport will considerably increase the computing time and memory. For example, a rectangular elementary cell with $l = 3h$ would increase the number of transitions in the sediment transport model by a factor of $3^2 = 9$. Beyond the proportional increase in calculation time, modifying the elementary cell ratio would also reduce the evolution rate of dune systems requiring even longer calculation times to reach steady-state configurations. In any case, changing the resolution of fluid flow lattice will results in another model that will require to:

- determine the length and time scales, $l_0$ and $t_0$ from a linear stability analysis using the most unstable wavelength, the mean flow strength and the corresponding saturated sand flux (*Narteau et al., 2009*; *Lü et al., 2021*).
- validate on the steady-state dune pattern that the shear stress and the associated sediment transport rate are representative of the dune elevation profile (*Zhang et al., 2010*).
- validate that the numerical approach reproduce dunes that (i) grow in height, (ii) migrate (propagation perpendicular to crest), and (iii) elongate (dune growth in a direction parallel to the crestline according to the wind regime and the sand availability) (*Zhang et al., 2012; Lü et al., 2017; Rozier et al., 2019; Courrech du Pont et al., 2024*).

This procedure must be identical for all cellular automaton dune models. To the best of our knowledge, this endeavor has only been accomplished for the ReSCAL dune model employed in this manuscript.
In conclusion, the idea of modifying the cell aspect-ratio of the model to validate behavior as subtle as the rates of repulsion and attraction is a great numerical prospect. Nevertheless, we estimate this work represents a considerable amount of work that is not compatible with the timing of a review and would neither impact the realism of the model nor the physical mechanism investigated in this paper.

***Specific comments***

☑ Lines 34-38 "*Instead, another set…*". Génois et al., 2013 presents an agent-based model of barchan systems in which there is no consideration of the flow and so it is not accurate to say that the work "indirectly highlights the long-range flow-induced interactions". Long-range interactions in that work occur only through the sand-flux exchange/shadowing i.e. an upwind dune prevents a downwind one from absorbing incoming flux. This has nothing to do with the fluid flow – it is a sediment-induced interaction.

→ We agree with this comment and removed the reference "Genois et al., 2013"

☑ Lines 39-42 "*Under unidirectional flows…*". It is not yet established whether this shearing at a long-distance takes place in aeolian dunes or solely in subaqueous experiments and highly turbulent numerical simulations. Some authors have claimed that in aeolian settings it is the case that it results from increased erosion induced by an upwind dune (e.g. Bourke, 2010) but others (e.g Elbelrhiti, 2012) have instead suggested that is it due to sand flux absorption and not due to erosion. It is worth mentioning that this debate for aeolian dunes has not been resolved.

→ We agree with this comment and reformulate this sentence now mentioning this alternative process. Suggested references have been added.

☑ Line 126 "*To prevent collisions…*". Presumably this means that below this initial separation collisions do occur. Since collisions are observed in nature, does this mean that dunes which collide must have formed closer to each other than this distance?

→ We add clarifications in this sentence in the line of this comment.

☑ Line 127 "*Except for the simulations shown in Figure 1, we assume a symmetric bidirectional flow regime with a period of flow reorientation. During each flow period, two flows of the same strength and duration blow alternately, so that there is zero resultant transport on the non-erodible bed away from any topography.*". Many linear dunes e.g. those shown in figure 1 form with an angular separation of less than 180 degrees. If the 2D bedforms shown in this work are taken to be cross-sections of these linear dunes then that would mean that different parts of the dunes would be interacting in the primary and secondary modes (since there is also a longitudinal component to the wind). How is this likely to affect the results presented here?

Line 158 "*Under these conditions, dunes are alternately located in the wake of the other dune and their resultant migration rate should be zero, except as a result of stochastic fluctuations.*"
Again, if the angular separation of the wind modes is less than 180 degrees different cross-sections of the dunes will be interacting under each mode.

→ In this paper, in order to work under optimal conditions that eliminate longitudinal flow components, we work in 2D with perfectly symmetric reversing flows. Considering this symmetric flow configuration, dunes are alternately located in the wake of the other dune with identical dune topography and spacing, and the relative migration rate of dunes should be zero. These specific conditions allow us to deconvolute the relative migration rate between dunes induced by the wake flow perturbations. Using this simplified set-up, we demonstrated that the reversing dune shape - i.e. its aspect ratio - controls the spatial structure of the turbulent wake, the distribution of the bed shear stress and the resulting steady-state spacing between dunes.
However, in nature, most linear dune fields form under bidirectional winds with a divergence angle smaller than 180° and transport ratio different than 1. Therefore, depending on the wind orientation, the apparent dune aspect ratio and interdune distance vary, modifying the steady-state spacing and the attraction/repulsion state of dunes. Then, compared to the optimal 2D conditions and the symmetric reversing winds used here, natural wind regimes introduce an additional level of complexity into the long-range flow-induced interactions between dunes, and the potential selection of linear dune wavelength. Further investigation is required to ascertain whether this wavelength is affected by the dominant wind (longer distance), the minor wind (shorter distance), or a combination of both (see Section 4.3).

☑ Line 154 "*Regardless of the initial spacing, dune pairs eventually reach the same equilibrium distance, where both dunes migrate at the same rate as a single dune under identical flow conditions.*" Provided that the initial spacing is above a predetermined length to avoid collisions.

→ We agree with this comment and clarify the sentence.

☑ Line 226 "*Wake flow perturbations provide an elementary mechanism for the development of a characteristic wavelength in dune fields with non-erodible interdune areas*"
What are the characteristic wavelengths in barchan dune fields which have non-erodible interdune

areas? If it is the flow then surely there would be a coupling between the size of the dunes and their spacing but this has been shown not to be the case in Durán et al. *Granular Matter* (2009)

→ Under unidirectional wind regime, the organization of barchan dune fields essentially relies on calving and collision processes due to the negative dependence of migration rates to dune size (Hersen, 2002; Duran et al., 2009; Worman et al., 2013; Robson & Baas, 2023). We here clarify that wake flow perturbations provide an elementary mechanism for the development of a characteristic wavelength in dune fields with non-erodible interdune areas submitted to bidirectional winds, where the long-range flow-induced effect on dune migration significantly modifies the dune migration rate. We therefore specify that is not the predominant controlling mechanism in barchan dune fields, which form and migrate (at high velocity) under unidirectional wind regime. We add some clarifications in the revised version of this article at the beginning of Section 4.3.

☑ Line 290 "*For example, before collision, an impacted dune should reduce (attraction) and then increase its migration rate (repulsion) as a smaller impacting dune is approaching (Fig. 8a).*"
Why then do we see collisions between barchans? Is it simply that the fact that you have a three-dimensional dune rather than a two-dimensional cross-section changes the whole behavior, in which case why should the results of two-dimensional simulations matter for studying barchans?

→ This comment and its response are in the line of the previous comment.
In this paper, we show that, under unidirectional wind regime, the wake flow perturbations induced by the upstream dune generate an acceleration or a deceleration ranging between 1% to 3% of the dune migration rate, therefore representing a very minor component of dune dynamics. Conversely, under a perfectly symmetric reversing wind regime when the resultant migration rate of dunes should be zero, these wake flow perturbations generate an acceleration superior to dune migration rate.
In barchan dune fields, which mostly develop under unidirectional wind regime, dune interactions essentially rely on collision processes resulting from differential migration rates according to their size: barchan migration rate is inversely proportional to their size. Considering that, wake flow perturbations should have a minor influence in barchan dune dynamics at a large scale and should not prevent their collision. However, at a shorter scale, this wake flow perturbations could be felt as a 2nd order component on barchan dynamics: just before collision, the impacted barchan could reduce and then increase its migration rate as a smaller impacting dune is approaching, favoring the ejection of a smaller barchan (as described in the Introduction of this article).
In the revised version of the article, we now clarify that our mechanism of wavelength selection by long-range flow-induced interactions cannot explain dune patterns in barchan fields, even if - at a second order - it can explain barchan behavior during collision processes. (see Section 4.3)

☑ Line 309 "*However, the results of the numerical model cannot yet predict the aspect ratio and wavelength observed within giant dune fields, where other processes related to collision, coarsening, superimposed bedforms or flow depth are likely to play a role in selecting the interdune distance*"
This is a really important point and could be emphasized further.

→ We agree with this comment. Therefore, we clarify in the introduction of the manuscript that our mechanism is not applicable to "giant type" of linear dunes formed under bidirectional wind regimes and we emphasized further this point in the section 4.3.

**Reviewer #2**

**General comments:**

☑ *Verify the absence of sediment exchanges between dunes*

Can you verify with the cross-sectional area of the dunes that there is no between-dune transport influencing the result? It's not clear how an initial spacing of 120 l0 is a sufficient guarantee.

→ Following an examination of the various sizes employed in this article ($S/l_0^2$ = {1000,1500,2000}), it can be confirmed that an initial interdune spacing of 120 $l_0$ is sufficient to prevent the exchange of sediment between dunes of 1000 $l_0^2$, which correspond to the dunes presented in Figures 4 and 5 of the article. Conversely, initial interdune spacings of 130 and 140 $l_0$ are necessary for dunes of 1500 and 2000 $l_0^2$, respectively. This will be explained in section 2.2, but does not affect the results presented in Figures 2 and 6, since initial interdune spacings greater than these thresholds are used.

☑ *How does the height of the domain influence the results ?*

How do your results depend on the height of the domain? 100 $l_0$ is about 1/3 of the dune height for the S=2000 $l_0^2$ case (as per Figure 2) – there is almost certainly 'pinching' of the flow over the dunes in this case. For lower S values, maybe the pinching is less important, but the varying levels of flow pinching between the dune crest and the domain top (akin to fluvial dunes), and it's role in influencing the stress across the dune stosses may make these results deviate from what could be observed in real aeolian dune systems. I think it's especially important to test this and discuss it since most of the results in the paper depend on it.

→ Indeed, as with all other parameters related to flow properties (i.e. flow strength, duration and direction), the flow depth (or domain height) impacts the equilibrium distance between dunes selected by long-range and flow-induced interactions, which has already been mentioned by *Andreotti et al. (2009)* and *Gao et al. (2015)*. Based on simulations that are not presented in this paper (with flow depth up to 400 $l_0$), we observe that higher flow depths - therefore reducing pitching effect between domain top and dune crest - does not remove the mechanism of repulsion/attraction between dunes. Indeed, we observe a positive dependence between the domain height and the equilibrium distance selected by attraction/repulsion mechanisms. However, further investigations need to be done in order to quantify this dependence.
Moreover, the domain height of 100 $l_0$ was also chosen to limit the computational cost related to the fluid flow model. Indeed, increasing the height of the domain would proportionally increase the calculation time for each simulation, considering each simulation consists of at least $2 \times 10^3$ flow periods. Additionally, this would also require an increase in the flow stabilization time ($10^4$ iterations of the lattice gas cellular automaton for a domain 100 $l_0$ in height) before each wind reversal.
Anyway, we agree with this comment and recognise that the domain height is a parameter that needs to be discussed further in the revised version of this article.

☑ *How are the aspect and shape ratios of simulated dunes similar to natural dunes?*

I think it would be worth mentioning if the simulated dune aspect and shape ratios are comparable to observed dunes in nature, to give more weight to the results shown here.

→ The typical slope of linear aeolian dunes on Earth, such as longitudinal or seif dunes, generally ranges from 10 to 15° on the stoss side (i.e. aspect ratio, *H/L* ~ 0.2), while the lee side or slip face slip face have slopes close to the angle of repose for dry sand around 30-35° (i.e. *H/L* ~ 0.6) (e.g. Lancaster, 1988;

Bristow et al., 2000). As mentioned in Section 2.1.5, $\tau_1/\tau_0 = 20$, $l_0 = 0.5$ m and $t_0 = 8.0 \times 10^{-4}$ yr are typical parameters for terrestrial aeolian dunes. In Figure 6, for $\tau_1/\tau_0 = 20$, the aspect ratio of simulated dunes typically ranges between 0.2 and 0.6, making simulated dunes comparable to natural ones. This point is now explicitly mentioned at the end of Section 3.4.

**Specific comments:**

☑ Line 20: Flow perturbation by aeolian dunes over long distances (i.e., beyond pair-wise interaction, at the dune-field scale) has been observed if the authors want a reference: https://doi.org/10.1029/2020GL088773

→ Reference has been added.

☑ Line 28: This sentence is true for dunes with a sufficiently steep lee side, it might be worth qualifying that a separation bubble does not form for all dunes.

→ We agree with this comment and a precision has been added.

☑ Line 45: "In other" should be "Another".

→ Words have been changed.

☑ Line 109: Controlling flow strength indirectly by $\tau_1$ is ok, but can you confirm that when you vary it, you also vary $\tau_2$ commensurately? A little more explanation of how this isn't quite the same as explicitly changing flow strength could be useful, e.g., non-linearity of flux-speed relationship, implicit Re-invariance of separation bubble size and speed-up effect, etc.

→ Since $(\tau_2-\tau_1)/\tau_0 = 100$, as it is mentioned in section 2.1.4, when $\tau_1$ varies $\tau_2$ also varies commensurately. As mentioned in *Narteau et al. (2009, Fig. 7)*, the minimum threshold $\tau_1$ for motion inception might reflect the average shear velocity ratio $u_*/u_{th}$, and ultimately the strength of the flow. Indeed, the saturated flux on a flat sand bed in the ReSCAL dune model continuously decreases with respect to the internal time unit $t_0$ when $\tau_1$ increases. Therefore, a high value of $\tau_1$ should be associated with a low wind speed. We add precision on this point in section 2.1.4 in the revised version of the manuscript.

☑ Line 146: Is $\langle\tau_{flat}\rangle$ measured on the flat topography of an identical domain with no dunes, or on flat topography in a domain with dunes in it? If it's the latter, then it should be clear what the condition for "away from any topography" is.

→ $\langle\tau_{flat}\rangle$ is the mean basal shear stress averaged over $150\,l\_0$ on a flat sand bed away from any topography. This long-range averaging in simulations with dunes is the same as on a flat bed. Due to the ergodicity of the flow within the model, time averaging at a point far from any topography over a very long period of time would give the same value. Precisions have been added at the end of section 2.2 in the revised version of the manuscript.

☑ Figure 2: "Unidirectionnal" typo in top left plot title. X-axis label on panel b says "x-ct", while panel f this is omitted.".

→ "Unidirectionnal" has been changed into "unidirectional" in Figure 2.
X-axis label on f is good since dunes in Figures c-f are not shifted, unlike those in Figures a-b.

☑ Figure 3: It would be useful to know what timestep in which experiment panels a and b correspond to, and if it is currently repulsing, attracting, or in equilibrium. In panel b, where does the dispersion band come from? Is it from the 10 random seeds or from multiple timesteps? Panels d and e aren't labeled.

→ Precisions have been added in the figure caption and panels "d" and "e" have been labeled.

☑ Line 172: This sentence is critical, yet I don't think it's reflected that well in Figure 3b. Here it says that downwind of a dune, once the flow has reattached, the stress is higher than if the dune wasn't there. That is only shown for the downwind dune, not the upwind dune – it's probably worth mentioning that when referencing the figure here.

→ We agree with this comment and the sentence has been clarified. Moreover, we modify Figure 3 in order to more clearly illustrate how the flow over a single dune impacts the distribution of the basal shear stress downstream.

☑ Line 174: Since $\Delta\langle\tau_s\rangle$ is normalised by $\langle\tau_{flat}\rangle$ I think maybe it's useful to put a \hat{} over it so it's clear it's normalised. I know they are all already non-dimensional but this one doubly so. Readers may take the quantities of ~1e-1 on the y-axis of Fig 3e literally.

→ We agree with this comment and a hat has been added to $\Delta\langle\tau_s\rangle$ (here and elsewhere) in order to avoid any confusion for the readers.

☑ Figure 5: caption for panel a should say "shape ratio (bottom)". The panel b is really nice, I wonder if it's possible to draw an analogy to pair-wise interaction potentials (e.g., Lennard-Jones) – especially since at even shorter ranges (i.e., in the recirculation bubble) the curve will be attractive again, right? But for panel b, I still do not think that it's demonstrated anywhere how the long-range attractive regime can be seen in the surface stress profile downwind of a dune. Shouldn't you see effectively a vertically flipped version of this curve in the \tau profile downwind of an isolated single dune? It would be great to see that – since in Figure 3b that attractive regime is missing.

→ Modifications have been done in the caption of Fig. 5
We agree with the comment of the reviewer. We modify the Figure 3 and add additional curves illustrating the distribution of the basal shear stress downstream of a single dune, in order to show - not only in Figure 5 - the origin of the long-range attractive regime where $\langle\tau_s\rangle < \langle\tau_{flat}\rangle$. Precisions have also been added in the body text (section 4.1) about the two additional unstable equilibrium distances that should result from this oscillatory behavior.

☑ Figure 6: It would be very useful to see if there's any way of collapsing these curves onto a master curve given the control parameters – it may not be simple though.
Also, why is the control parameter set sparse? E.g., reversing flow \tau_1/\tau_0=30 data, and (\tau_1/\tau_0<30)&(S<2e+3l_0^2) data, are not shown.
Now also, it should be stated when exactly the aspect ratio shown on this graph is being measured – is this the time-averaged aspect ratio, or at the flow reversal, or at t/\DeltaT=1/4, etc.? Figure 5a demonstrates well that this specificity is required.

→ We agree with this comment and the first idea regarding this figure was to build an abacus depending on controlling parameters. However, this proved to be difficult to understand and too complex for realization.
Dataset under reversing flow regimes with $\tau_1/\tau_0 = 30$ have not been computed. Therefore, to avoid any confusion for the readers, we removed the related points under unidirectional flow.

While the set of parameters could indeed be more dense, the aim of this study is basically to examine the quantitative dependency on dune size and threshold shear stress (i.e. flow strength) on the equilibrium distance selected by long-range flow-induced interactions.

In the revised version of the manuscript, we specify when the dune spacing and aspect ratio measurements are made.

☑ Line 246: isn't this perturbation decay the simplest explanation for the slope change in Figure 3e?

→ We agree with this comment and notice that this point is well explained in the next part of this section.

☑ Line 253: The sentence "Our model…" should probably come with an explanation of why you think your model is not accurate enough – i.e., provide its limitations.

→ We agree with this comment and we now more clearly explained the limitations of our lattice gas model for fluid flow, especially compared with computational fluid dynamic models.

☑ Line 262: For the sentence "This could be…", I feel like this could've been tested using the simulations. You did not quantify the lengthscales of the recirculation bubble or of the upstream shear-stress drop, or show how they depend on dune shape or size.

→ Compared with computational fluid dynamic models that solve the Navier-Stokes equations, our discrete lattice gas model for fluid flow may not have sufficient accuracy to evaluate in greater detail the turbulence intensity in the various flow regions observed in the wake of dunes. It is possible to reproduce the recirculation zone with a length of a few dune heights, but it is difficult to identify and calculate the geometry of secondary flow regions in the reattachment area, therefore not enabling to study their dependance on dune size and shape. This is why we recommend future work using laboratory experiments and advanced numerical methods to quantitatively investigate this point.

☑ Section 4.3: This is an interesting discussion, but I think it's definitely worth noting earlier on (rather than at the end) that we expect from various data (ReSCAL simulations included) that dune spacing during dune field formation is predominantly controlled by a pattern coarsening process, even when resultant fluxes are small. A 'steady state' dune field—or one where coarsening rate is low enough that pair-wise potential between similarly-sized dunes is the dominant control on dune spacing—may not be very common on Earth since climate changes faster than large dunes can keep up with (e.g., https://doi.org/10.2110/jsr.2018.55 and https://doi.org/10.1130/G50837.1). Overall though I believe it's a provocative and balanced section.

→ We agree with the comment and now further discuss in section 4.3 (as well as earlier in the manuscript, at the beginning of the introduction) under which conditions this spacing-selection mechanism is thought to occur in aeolian dune fields.

**Community Comment - Yuanwei Lin**

☑ Fig. 1d & Line 45-49: I recommend further emphasizing that these materials originate from the flume experiments. Also, this figure looks so great that I suspect it may be the result of post-processing water flume experiment data. Therefore, I suggest including the post-processing methods and procedures for clarification.

→ Precisions mentioning the "flume" origin of experiments have been added in the body text of the introduction as well as in the caption of Fig. 1d. Clarifications have also been added in the figure caption regarding the post-processing of flume experiment snapshots.

☑ Line 61: ...or barchans (Zhang et al., 2014; Lin et al., 2024).

→ References have been added.

☑ Line 64: One sentence (To eliminate...) is obviously insufficient to explain the reason for conducting only 2D simulations. The authors may provide further clarification on this issue by referring Jarvis et al. (2023).

→ We concur with this comment and have included further clarifications and references in the Introduction regarding the differences between 2D and 3D simulations, especially regarding the dune interactions, turbulence dynamics and pattern coarsening processes.

☑ Line 119-121: It is suggested to mention examples of physical scaling in underwater experiments in this sentence by referring to Javis et al. (2023).

→ For the estimation of the length and time scales of the model according to the value of the threshold shear stress for the motion inception, we prefer to keep the value provided by Zhang et al. (2014) for subaqueous experiments. The reason is that in the experiments of Jarvis, the most unstable wavelength lambda max was computed from a formula and not directly estimated from laboratory experiments, giving some uncertainties to the estimation of time scales. Note that there is not a big difference between the values provided by Jarvis et al., 2023 ($l_0 = 2.27 \times 10^{-3}$ m and $t_0 = 9.74 \times 10^{-3}$ s $= 3.08 \times 10^{-10}$ yr) and those in Zhang et al., 2014 ($l_0 = 0.5 \times 10^{-3}$ m and $t_0 = 1.6 \times 10^{-10}$ yr), which relies mainly on the difference in grainsize between the model and the experiments.

☑ Line 127-129: Since previous publications have studied long-range dune interactions under unidirectional flow (Bacik et al., 2020; He et al., 2023), I was initially confused by the setting of bidirectional flow when I read this sentence. After reading this manuscript several times, I understood that the symmetrical bidirectional flow here corresponds to the evaluation of linear dunes under multi-directional flow regimes, as illustrated in Fig. 1c and d. Therefore, I suggest bridging the setting of bidirectional (or multi-directional) flow with Fig. 1c and d through a few explanatory sentences to enhance readers' understanding.

→ We agree that this point was probably not clear enough in the first version of the article. In the revised version, we now more clearly explain why we work under symmetric reversing flow regimes at the end of the introduction.

☑ Line 129-131: I would like to understand the reason behind implementing 10^4 iterations after each flow reversal and why this specific number 10^4 was selected. Additionally, it is worth explaining whether this parameter setting has been validated through pre-simulation testing.

→ Since the mechanism proposed in this paper to explain the existence of an equilibrium distance between the dunes relies essentially on the perturbation of the turbulent flow downstream of the dune, it was necessary to ensure that the turbulent structures were well developed (i.e. recirculation bubble and basal shear stress drop on the lee side) at each flow reversal. We tested several flow stabilization times ($10^2$, $10^3$ and $10^4$ $t_0$) on static dunes and it turned out that with a domain 1000 $l_0$ long and 100 $l_0$ high, the required stabilization time was $10^4$ $t_0$.

☑ Line 168-174 & Figs. 3a & b: To my knowledge, this concept and similar figures have been presented many times in previous papers (Wang et al., 2018; Cai et al., 2021), making it unnecessary to emphasize them further in the current manuscript.

→ We agree that this concept and similar observations have already been made by tens of laboratory experiments, numerical simulations and field measurements, listed in the second paragraph of the introduction (including references proposed in this comment). However, given the mechanism proposed in this paper is intimately related to the structure of the turbulent wake in the lee of dunes, we assume the choice to keep this description in our Results.

☑ Line 216-217: According to this sentence and Fig. 6, the equilibrium distance is affected by three factors: dune size, flow strength and flow reorientation periods. There is an open question regarding whether flow strength and reorientation periods can be integrated into single variable termed flow conditions, i.e., flow intensity with positive and negative values indicating flow reversals.
The relationships depicted in Figure 6 are highly consistent. Consequently, I anticipate the development of an intriguing unifying law and a compelling figure featuring dimensionless λ_D on the y-axis and a parameter combination of flow conditions and dune size on the x-axis. Certainly, I acknowledge that establishing this law in the short term is challenging and requires further discussion. However, the authors could consider including this idea as part of the prospects in the conclusion of the current manuscript.

→ We agree with this comment and add a sentence about this perspective in the Conclusion of the revised paper.

☑ Line 225: "underlining the long-term morphological impact of crest reversal on dune shapes." It is recommended to supplemented few sentences to explain in detail the impact of crest reversal on dune shapes.

→ We agree with this comment and we add a detailed description to explain the impact of crest reversal on dune shapes at the end of Section 3.4.

---

## Referee Report (RR1)

My comments on the original manuscript were focused on two main perceived issues: the use of a square lattice and the achievement of the angle of repose. On the first of these points the authors have given a detailed explanation as to:

1) Why they believe that changing the (height) aspect ratio of grid cells would not have a significant impact on the results
2) The amount of additional work that would be required to do so.

On the first of these points, I am willing to accept the authors' argument that the basal shear stress distribution (which is the key determinant of migration rate) would not be significantly impacted by changing the aspect ratio. The authors do concede that there would be some changes "but may require to change the distance over which the shear stress is computed" but that the fundamental presence of a equilibrium distance would not be changed. This may well be the case, and given the authors' explanation of the amount of additional work that would be required to change the aspect ratio, I am satisfied with this.

On the second main concern I raised, pertaining to the angle of repose, the authors have included additional text explaining that the local slope is calculated over five cells which does resolve the issue I had. Although I still have some queries listed below.

Aside from these, the authors have taken on board all of my other comments and made appropriate changes to the manuscript. The article has been improved under the revision remains well-written and presented and I believe it will interest many in the community. Therefore, with some minor changes, I would recommend publication.

Specific suggestions:

Line 99 – The authors have included additional description of how the slope is calculated over more than one cell – thus enabling a more accurate angle of repose to be imposed. The authors state that this was done in Zhang et al. (2014) however I cannot find this in the text of that article (perhaps I am missing something?). The other reference (Gao et al. (2016)) does include a specific statement of this rule in its Supplementary Material. I don't believe that it is stated in either of the sources why the choice was made to calculate over this particular number of cells – but this is only a very minor point.

The methods section should include a short version (a sentence or two) of the argument that the aspect ratio of cells should not impact the mechanisms this work describes.

---

## Author Response (AR2)

REVIEW - EGUSPHERE-2024-1634

**Equilibrium distance from long-range dune interactions**

Earth Surface Dynamics

⇒ **General reply:**

Dear Editor,

We thank the two anonymous invited reviewers for the second round of reviews, and their positive feedback on the current revised version of the manuscript.

Two minor modifications were however suggested. They have been applied and they are explained below:

- *Check the citation of Zhang et al. (2014) for the implementation of the angle of repose.*
  In the reference Zhang et al. (2014), it is clearly stated that the implementation of an avalanche module in ReSCAL based on a diffusion with threshold mechanism improves the sedimentary patterns associated with gravity-driven granular flows
  → See the following paragraph in Zhang et al. (2014) at page 454 "In the model, we consider a so-called diffusion process [...] In order to ensure that the slope is never larger than $\theta c = 35°$, $\Lambda ava \gg \Lambda e$ in all the numerical simulations described below"
  We also precise that further details are provided in the Supplementary Material S1 of Gao et al. (2016).

- *Include a short summary of the author's response comments on the aspect ratio of cells in the manuscript.*
  Following this comment, we have added two sentences synthesizing the discussion on the non-influence of the elementary cell aspect ratio, at the end of Section 2.1.4.

Kind regards,
The author team.